# CRIg, a tissue-resident macrophage specific immune checkpoint molecule, promotes immunological tolerance in NOD mice, via a dual role in effector and regulatory T cells

Xiaomei Yuan[1], Bi-Huei Yang[1], Yi Dong[1], Asami Yamamura[2], Wenxian Fu[1,2,3,4]*

[1]Pediatric Diabetes Research Center, Department of Pediatrics, University of California, San Diego, United States; [2]Biomedical Sciences Graduate Program, University of California, San Diego, United States; [3]Institute for Diabetes and Metabolic Health, University of California, San Diego, United States; [4]Moores Cancer Center, University of California, San Diego, United States

**Abstract** How tissue-resident macrophages (TRM) impact adaptive immune responses remains poorly understood. We report novel mechanisms by which TRMs regulate T cell activities at tissue sites. These mechanisms are mediated by the complement receptor of immunoglobulin family (CRIg). Using animal models for autoimmune type 1 diabetes (T1D), we found that CRIg[+] TRMs formed a protective barrier surrounding pancreatic islets. Genetic ablation of CRIg exacerbated islet inflammation and local T cell activation. CRIg exhibited a dual function of attenuating early T cell activation and promoting the differentiation of Foxp3[+] regulatory (Treg) cells. More importantly, CRIg stabilized the expression of Foxp3 in Treg cells, by enhancing their responsiveness to interleukin-2. The expression of CRIg in TRMs was postnatally regulated by gut microbial signals and metabolites. Thus, environmental cues instruct TRMs to express CRIg, which functions as an immune checkpoint molecule to regulate adaptive immunity and promote immune tolerance.

DOI: https://doi.org/10.7554/eLife.29540.001

*For correspondence:
w3fu@ucsd.edu

**Competing interests:** The authors declare that no competing interests exist.

## Introduction

Most complex diseases are associated with a tissue-specific chronic inflammation. Excessive inflammation has deleterious consequences, characterized by tissue damage and disease onset (*Medzhitov, 2008*; *Grivennikov et al., 2010*; *Olefsky and Glass, 2010*). Understanding cellular and molecular mechanisms that can keep inflammation in check will have important therapeutic implications in autoimmune and inflammatory diseases. Macrophages, especially tissue-resident populations (TRMs) represent a key constituent of the innate immune system in orchestrating inflammation and influencing the functions of other immune and non-immune cells (*Medzhitov, 2008*; *Davies et al., 2013*). In concert with other innate immune cells (e.g., mast cells), macrophages sense environmental stimuli and subsequently instruct adaptive immunity (primarily T cells), together to elicit the processes of host defense and tissue repair (*Iwasaki and Medzhitov, 2015*).

Under autoimmune conditions, self-reactive T cells accumulate at tissue sites and lead to tissue destruction. T cell activities need to be restrained to maintain immunological tolerance (*Sharpe and Freeman, 2002*). However, how these autoimmune T cell responses are influenced by tissue microenvironment remains poorly understood. We hypothesize that macrophages, especially TRMs may

fulfill such a role to modulate tissue T cell activities. Multiple mechanisms may be exploited by these TRMs to regulate T cell activities. First, these TRMs express multiple coinhibitory molecules of B7/CD28 and TNFR families, such as PD-L1, PD-1 and TIM-4 (refs [*Kuang et al., 2009*; *Gordon et al., 2017*; *Thornley et al., 2014*]). Second, TRMs may regulate the generation, stability, and function of Foxp3+ regulatory T (Treg) cells. This notion is supported by a few studies (*Denning et al., 2007*; *Soroosh et al., 2013*; *Haribhai et al., 2016*). These studies have demonstrated that macrophages residing in the lung (*Soroosh et al., 2013*), or intestinal lamina propria (*Denning et al., 2007*; *Haribhai et al., 2016*) can promote the generation of Treg cells. These two mechanisms are not mutually exclusive, because certain coinhibitory molecules (e.g., PD-L1) have been found to promote the generation of Treg cells (*Francisco et al., 2009*; *Wang et al., 2008*; *Amarnath et al., 2011*).

Recently, a new subset of TRMs has been identified (*Helmy et al., 2006*; *Vogt et al., 2006*; *Fu et al., 2012*). These macrophages are phenotypically characterized as CD11b+ F4/80hi CRIg+. CRIg+ macrophages can be detected in peritoneal cavity and most digestive organs, including pancreas, liver, small and large intestines (*Helmy et al., 2006*; *Vogt et al., 2006*; *Fu et al., 2012*; *Gautier et al., 2012*). Macrophages from lymphoid organs do not express CRIg. CRIg is also absent in other immune cells, including T cells, B cells, NK cells, dendritic cells, monocytes, and neutrophils (*Helmy et al., 2006*; *Vogt et al., 2006*). Thus, the expression of CRIg is restricted to TRMs. CRIg (gene symbol: *Vsig4*) was previously reported as a complement receptor (*Helmy et al., 2006*). It binds to complement components C3b, iC3b and C3c, and is involved in the phagocytosis of microbes (*Helmy et al., 2006*; *Wiesmann et al., 2006*). However, further studies including ours, have revealed that CRIg can function as a coinhibitory molecule of the B7/CD28 superfamily (*Vogt et al., 2006*; *Fu et al., 2012*). In this regard, CRIg potently suppresses T cell proliferation and cytokine production (*Vogt et al., 2006*; *Fu et al., 2012*). Given that CRIg is exclusively expressed by TRMs, we propose that CRIg mediated signaling represents a tissue-derived mechanism by which resident macrophages regulate tissue homeostasis and inflammatory processes.

In autoimmune type 1 diabetes (T1D), islet inflammation (insulitis) is exacerbated by the loss-of-balance between regulatory and pathogenic immune cells (*Bluestone et al., 2010*). However, not all individuals with insulitis proceed to become diabetic, suggesting that there exist mechanisms that can keep insulitis in check. We have previously found that the abundance of CRIg+ macrophages in the pancreas was negatively correlated with the risk of diabetes onset in non-obese diabetic (NOD) mice, the primary animal model for T1D. More importantly, in vivo administration of CRIg-Ig fusion protein suppressed diabetes in these mice (*Fu et al., 2012*). In this study, we attempted to understand how CRIg impacts T cell activities using mouse T1D as a primary model system. Our studies revealed a dual role of CRIg in regulating effector T cell proliferation and Treg differentiation and stability. Together, these two outcomes led to a reinforced immune tolerance in NOD mice. Furthermore, we investigated the role of gut microbiota and derived metabolites in regulating the expression of CRIg in TRMs. Together, these findings support a new model of immune regulation in tissue homeostasis whereby environmental cues instruct TRMs to express CRIg, which functions as an immune checkpoint molecule to regulate adaptive immunity and promote immune tolerance.

## Results

### CRIg+ TRMs form a protective barrier surrounding pancreatic islets to prevent autoimmune infiltration in NOD mice

CRIg is exclusively expressed in TRMs (*Helmy et al., 2006*; *Vogt et al., 2006*; *Fu et al., 2012*). CRIg+ macrophages can be detected in most digestive organs at steady state, including the pancreas, liver, colon and peritoneal cavity of both C57BL/6 (B6) and NOD mice (*Figure 1-figure supplement 1A, B* and data not shown). CRIg+ cells was also found in human pancreas (*Figure 1—figure supplement 1C*). There was no CRIg+ cell in the lung in both mouse strains (*Figure 1—figure supplement 1A and B* and data not shown). The expression of CRIg was not detected in lymphoid-lineage cells, including T cells, B cells and NK cells (*Figure 1—figure supplement 1D*). In the pancreas, CRIg+ TRMs were enriched at the capsular area of the islets (*Figure 1—figure supplement 1A*), exhibiting a perivascular distribution (*Fu et al., 2012*). We have previously found that the abundance of islet CRIg+ macrophages in NOD mice was negatively correlated with the risk of diabetes onset in these animals (*Fu et al., 2012*), suggesting a disease-suppressing role of CRIg+ TRMs in

T1D. However, the mechanisms remained undefined. Given the unique tissue distribution pattern of these cells, we examined the relationship between the severity of insulitis and the abundance of peri-islet CRIg[+] TRMs on a per-islet basis in pre-diabetic NOD mice. We found that intact islets (free from autoimmune infiltration) were associated with more abundant peri-islet CRIg[+] TRMs. In contrast, in insulitic islets, there was a lack of CRIg expression in TRMs, especially in those peri-islet areas succumbed to infiltration (*Figure 1A*). These results suggest that CRIg[+] TRMs may function as a local barrier to prevent immune infiltration into pancreatic islets. We proposed that CRIg molecule was critical for these TRMs to exhibit such a protective role in insulitis. To confirm the role of CRIg, we generated CRIg-null mutation onto a NOD background. Indeed, genetic ablation of CRIg

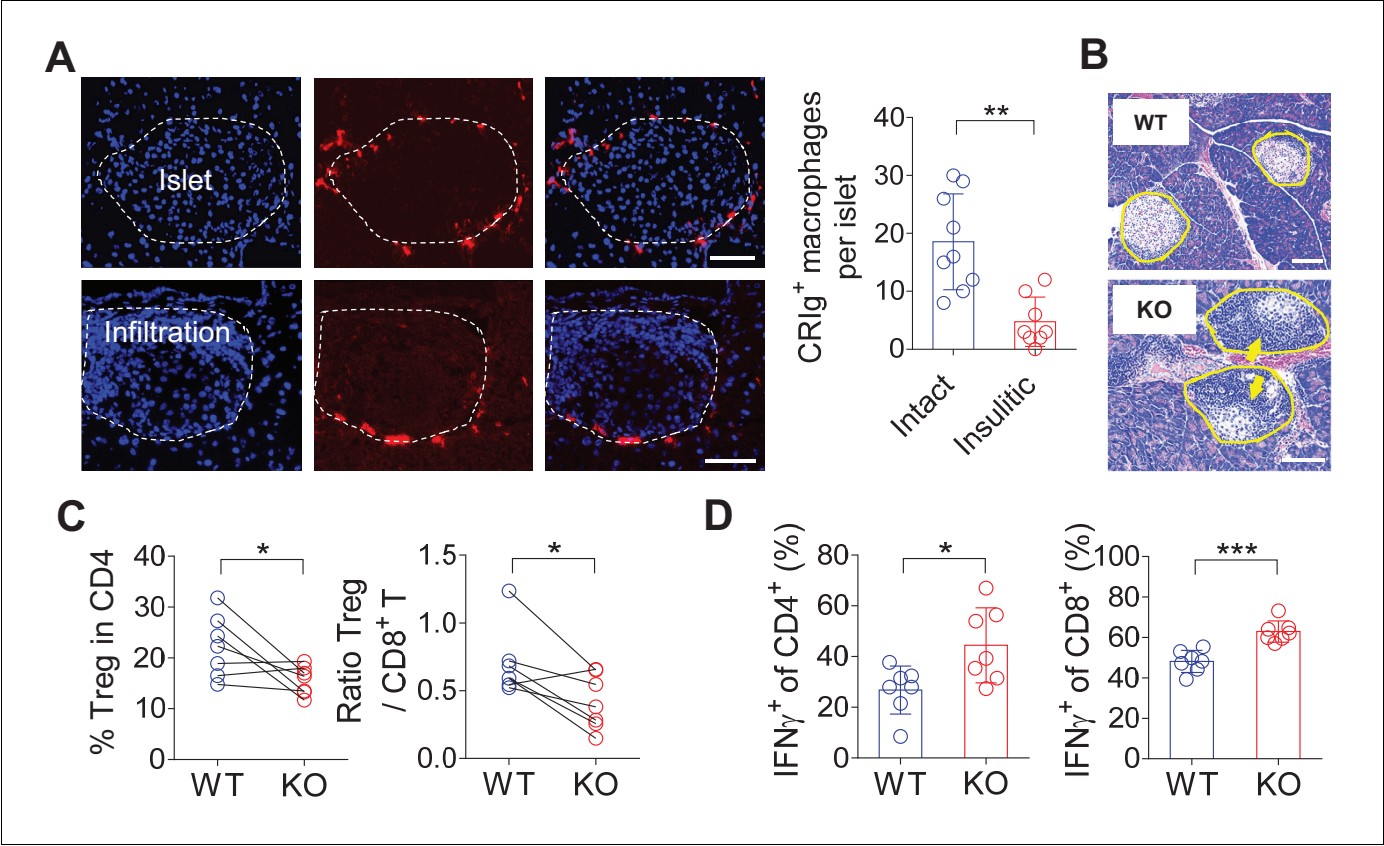

**Figure 1.** CRIg[+]TRMs form a protective barrier to prevent tissue autoimmune infiltration and activation. (**A**) The distribution of CRIg[+] TRMs within the pancreas. Immunostaining of pancreatic frozen sections of 10-week-old NOD mice. (left) Representative images depicting an intact (upper) and an insulitic (lower) islet, respectively. Red: CRIg; Blue: DAPI. The border of the islet was marked by dotted lines; immune infiltration was highlighted by higher density of DAPI[+] dots (n = 9 for intact and n = 8 for insulitic). Bar, 50 um. (right) The numbers of CRIg[+] TRMs per islet counted from immunostaining sections of pancreas as in (**A**). Islets were categorized into intact (free from infiltration) and insulitic groups. (**B**) The severity of insulitis in 10-week-old NOD and NOD/CRIg KO mice. Hematoxylin and eosin staining of pancreatic paraffin sections. Arrowheads depict massive immune infiltration. Bar, 50 um. (**C**) Flow cytometric analyses of digested pancreases of NOD/CRIg KO mice and littermate controls (n = 7 in each group). (Left) The percentages of pancreatic Treg cells in age-matched NOD and NOD/CRIg KO mice (mixed of both females and males). (Right) The ratio between Treg cells and CD8[+] T cells (n = 7 in each group). (**D**) The production of IFN-γ in CD4[+] Tconv and CD8[+] T cells. Pancreatic digestions were the same as in (**C**). Data are representative of three (**A, B**) or four (**C, D**) experiments. Student's *t*-test was used. *p<0.05; **p<0.01; ***p<0.001.

DOI: https://doi.org/10.7554/eLife.29540.002

The following figure supplements are available for figure 1:

**Figure supplement 1.** Tissue-distribution of CRIg[+] TRMs.
DOI: https://doi.org/10.7554/eLife.29540.003

**Figure supplement 2.** CRIg deficiency does not affect T cells in lymphoid organs.
DOI: https://doi.org/10.7554/eLife.29540.004

**Figure supplement 3.** Cell-cell contact between CRIg[+]TRMs and T cells in pancreas.
DOI: https://doi.org/10.7554/eLife.29540.005

significantly exacerbated the severity of insulitis, reflected by a massive infiltration of leukocytes in most islets (*Figure 1B*), supporting a critical role of CRIg for these CRIg$^+$ TRMs to suppress insulitis. This was consistent with our previously published observations showing that the lower the number of pancreatic CRIg$^+$ TRMs, the higher the incidence of diabetes in NOD mice (*Fu et al., 2012*).

We next asked whether CRIg deficiency in TRMs was correlated with altered compositions and activities of islet T cells. Due to the heterogeneities of insulitis in NOD mice, we calculated the ratios between regulatory and pathogenic cells in pancreatic islets to more accurately reflect how CRIg deficiency affected the severity of insulitis within each individual mouse. Comparing to age-matched wildtype NOD mice, CRIg knockout (KO) mice exhibited an altered balance between regulatory and effector T cells, reflected by: (i) The fraction of Treg cells was reduced in pancreatic islets (*Figure 1C*). Such a reduction of Treg abundance was not seen in lymphoid organs (*Figure 1—figure supplement 2*), supporting a tissue-specific immunoregulatory role of CRIg. (ii) The ratio between pancreatic Treg and CD8$^+$ T cells was also reduced in NOD/CRIg KO mice (*Figure 1C*). We next analyzed cytokine production in T cells isolated from NOD/CRIg KO and littermate control mice. Both CD4$^+$ and CD8$^+$ T cells isolated from the pancreatic islets, not lymphoid organs, of NOD/CRIg KO mice exhibited significantly increased production of IFN-γ (*Figure 1D*), which is a key pathogenic cytokine in T1D development (*Feuerer et al., 2009*). We next examined whether CRIg$^+$ TRMs interacted with T cells. Immunostaining of pre-diabetic islets (with ongoing insulitis) from NOD mice showed that there were cell clusters containing both CRIg$^+$ TRMs and T cells in close proximity, suggesting a cell-cell contact between CRIg$^+$ TRMs and T cells (*Figure 1—figure supplement 3*).

In summary, pancreatic CRIg$^+$ TRMs form a protective barrier surrounding the islets to prevent insulitis and impact local immuno-balance between effector and regulatory T cells. CRIg molecule plays a central role for these TRMs to exert their diabetes-suppressing effect.

## CRIg suppresses T cell proliferation by attenuating early T cell activation signaling

CRIg has been reported to suppress T cell activation and cytokine production (*Vogt et al., 2006*; *Fu et al., 2012*). However, how CRIg regulates T cell activities remains unknown. Consistent with previous reports (*Vogt et al., 2006*; *Fu et al., 2012*), we found that plate-coated CRIg-Ig, not control Ig, suppressed T cell proliferation in vitro (*Figure 2A*). We asked whether CRIg affected early activation phase, or late proliferation phase, of T cells. To answer this question, we isolated CD4$^+$ CD25$^-$ conventional T (Tconv) cells and cultured them in vitro for 3 days with anti-CD3/CD28. These cultured T cells were supplemented with plate-coated CRIg-Ig under two different conditions - either only the first 24 hours (hr), or only the late 48 hr, of the in vitro culture. We found that the presence of CRIg-Ig during the first 24 hr was sufficient to suppress T cell proliferation, to an extent similar to the condition where CRIg-Ig was present all time of the culture (*Figure 2A*). In contrast, when CRIg-Ig was only present during the last 48 hr, its suppressive effect on T cell proliferation was largely abolished (*Figure 2A*). Therefore, CRIg interferes early T cell activation. This notion was further supported by flow cytometric analysis of the expression of CD69 and CD25, two cell-surface markers depicting T cell activation. In a time-course assay, the majority of T cells exhibited an upregulation of both CD69 and CD25 within 17 hr. CRIg-Ig potently suppressed the upregulation of both CD69 and CD25 in T cells (*Figure 2B*). We then examined whether CRIg affected TCR signaling pathways. Using phosphorylation assays, we found that the presence of plate-bound CRIg-Ig suppressed the phosphorylation of ZAP70, ERK1/2, AKT and ribosomal protein S6 (*Figure 2C*). These data suggest that CRIg-Ig dampens several pathways that are crucial for early T cell activation, including the most proximal signaling complex (ZAP70), the ERK-MAPK pathway (ERK1/2, and S6) and the AKT-MTOR pathway (AKT, S6).

CRIg has also been reported as a complement receptor that binds to C3b, iC3b and C3c (*Helmy et al., 2006*). We asked whether complement components were involved in CRIg elicited T cell suppression. Using a specific monoclonal antibody (mAb) (clone 14G8) (ref. [*Gorgani et al., 2008*]) that can block the binding of CRIg to C3 components, we found that 14G8 did not abolish T cell suppression by CRIg, suggesting that complements were not involved in CRIg-mediated T cell suppression (*Figure 2—figure supplement 1A*). This notion was further supported by that CRIg can attenuate TCR activation in serum-free condition, which was deprived of complements (*Figure 2—figure supplement 1B*). In fact, CRIg-Ig mediated suppression of T cell proliferation was even enhanced by anti-CRIg (e.g., clone 14G8) (*Figure 2—figure supplement 1A*), or other clonotypes of

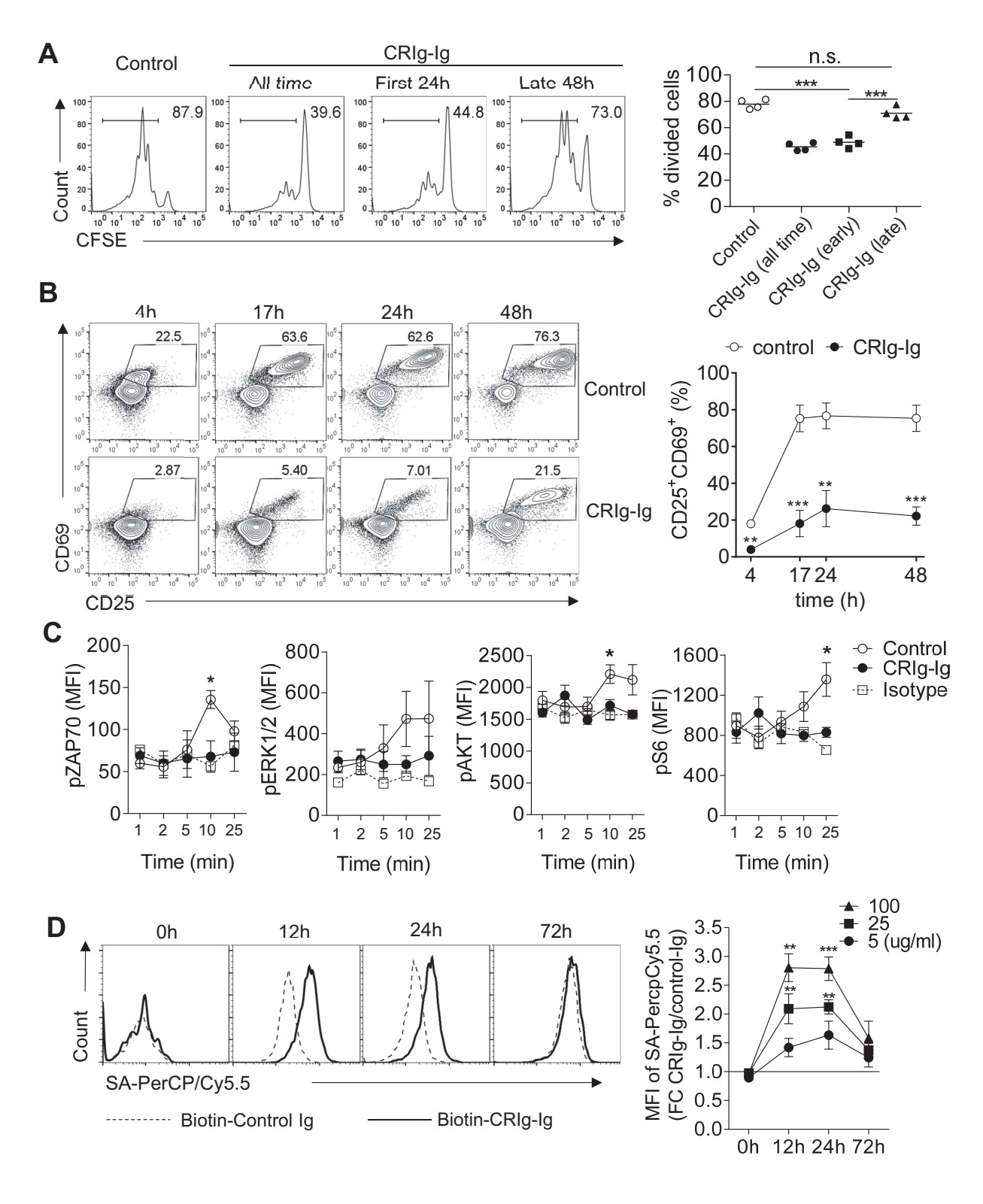

**Figure 2.** CRIg suppresses T cell activation. (**A**) CFSE-labeled CD4[+] CD25[−] Tconv cells were stimulated with anti-CD3/CD28. Plate-bound CRIg-Ig, or control Ig, was added either all time during T cell culture (3 d), or only the first 24 hr, or the late 48 hr. T cell proliferation was measured by CFSE dilution. (**B**) The expression of early T cell activation markers-CD69 and CD25 in cultured Tconv cells with plate-bound control Ig, or CRIg-Ig. (**C**) The phosphorylation of early T cell activation cascade proteins. CD4[+] CD25[−] Tconv cells were activated in vitro with anti-CD3/CD28 in the presence of plate-

*Figure 2 continued on next page*

Figure 2 continued

bound control Ig, or CRIg-Ig. (D) The binding of CRIg to activated T cells. CD4$^+$ CD25$^-$ Tconv cells were activated in vitro by anti-CD3/CD28 for various lengths of time (12 hr, 24 hr and 72 hr). At each time-point of T cell activation, various concentrations of biotinylated control Ig or CRIg-Ig were incubated with T cells at 37$^\circ$C for 1 hr. The binding of biotin-labeled Ig proteins was detected using streptavidin-conjugated antibody. (left) Representative histograms of SA-PercpCy5.5 MFIs depicting the binding intensities of biotinylated proteins (25 ug/ml) bound to T cells. (right) Statistical data showing fold-changes of SA-PercpCy5.5 MFIs between CRIg-Ig and control Ig at each time-point and each concentration. *P* values were calculated by comparing the binding intensities between biotin-CRIg-Ig and biotin-control-Ig. The data are representative from five (A), three (B), and four (C, D) experiments. Student's *t*-test was used. *p<0.05; **p<0.01; ***p<0.001, n.s., non-significant.

DOI: https://doi.org/10.7554/eLife.29540.006

The following figure supplements are available for figure 2:

**Figure supplement 1.** The suppressive effect of CRIg on T cells is complement-independent.

DOI: https://doi.org/10.7554/eLife.29540.007

**Figure supplement 2.** Plate-bound anti-CRIg mAb augments the effect of CRIg-Ig in T cells.

DOI: https://doi.org/10.7554/eLife.29540.008

**Figure supplement 3.** The binding of CRIg to activated Treg cells and differential suppression of CRIg-Ig for Tconv and Treg cells.

DOI: https://doi.org/10.7554/eLife.29540.009

**Figure supplement 4.** Ig fusion proteins of CTLA4, PD-1, VISTA, CD226 and TIGIT do not abolish the suppression of CRIg-Ig in T cells.

DOI: https://doi.org/10.7554/eLife.29540.010

anti-CRIg mAbs that do not interfere the binding of CRIg to C3 components (data not shown), suggesting a possible cross-linking effect by anti-CRIg mAbs on CRIg-Ig. In line with this, we found that soluble CRIg-Ig did not suppress T cell proliferation. In contrast, pre-coating the plates with anti-CRIg mAb (clone 14G8, or 17C9) enabled soluble CRIg-Ig to potently suppress T cell proliferation (*Figure 2—figure supplement 2*).

We then asked whether CRIg can directly bind to T cells. In this regard, isolated CD4$^+$ Tconv cells were activated in vitro with anti-CD3/CD28 and cultured for various lengths of time (12, 24, and 72 hr, respectively). At each stage of T cell activation, these cells were incubated with biotinylated CRIg-Ig or control Ig at 37°C for 1 hr, and the binding was measured using a streptavidin-conjugated secondary Ab. We found that CRIg can bind to activated T cells, in a dose-dependent manner (*Figure 2D*). Interestingly, we observed stronger binding of CRIg-Ig to T cells at their early, not late, stage of in vitro activation (12 hr and 24 hr versus 72 hr) (*Figure 2D*). This was consistent with the suppressive effect of CRIg-Ig on T cells (*Figure 2A*). CRIg-Ig can also bind to activated Treg cells. However, in any analyzed concentration of CRIg-Ig, or time-point, the binding intensity was lower comparing to that for Tconv cells (*Figure 2—figure supplement 3A*). Consistent with this, Treg cells were less sensitive to CRIg-mediated suppression (*Figure 2—figure supplement 3B*).

The extracellular domains of CRIg contain one IgV-type of immunoglobulin (*Helmy et al., 2006*; *Vogt et al., 2006*). Without knowing the cognate receptor(s) for CRIg, we tested whether CRIg could bind to any known coinhibitory molecules of immunoglobulin family. In a competition assay, CD4$^+$ Tconv cells were activated with anti-CD3/CD28 in the presence of plate-bound CRIg-Ig, or control Ig. Based on the availabilities of Ig fusion proteins, we tested CTLA-4, PD-1, VISTA, CD226 and TIGIT to assess whether the soluble forms of any of these proteins can block the effect of CRIg-Ig on T cell proliferation. None of these tested soluble proteins exhibited a blocking effect (*Figure 2—figure supplement 4*).

In summary, CRIg preferentially suppresses effector T cells via binding to a putative receptor in these cells at their early stage of activation and attenuating early TCR activation signaling.

## CRIg enhances iTreg differentiation in vitro and regulates pTreg development at tissue sites

B7/CD28 family coinhibitory molecules (e.g., PD-L1) have been reported to promote Treg differentiation (*Francisco et al., 2009*; *Wang et al., 2008*; *Amarnath et al., 2011*). We asked whether CRIg has a similar effect on Treg differentiation. We assessed this possibility in an in vitro TGF-β - induced Treg (iTreg) condition (*Chen et al., 2003*). Purified CD4$^+$ Foxp3(GFP)$^-$ Tconv cells from the spleen and LNs of NOD/Foxp3$^{GFP}$ mice (*Haribhai et al., 2007*) were cultured in the presence of anti-CD3/CD28, IL-2 and TGF-β to induce iTreg differentiation. As expected, TGF-β promoted iTreg differentiation in a dose-dependent manner. At any given dose of TGF-β, CRIg-Ig further enhanced the

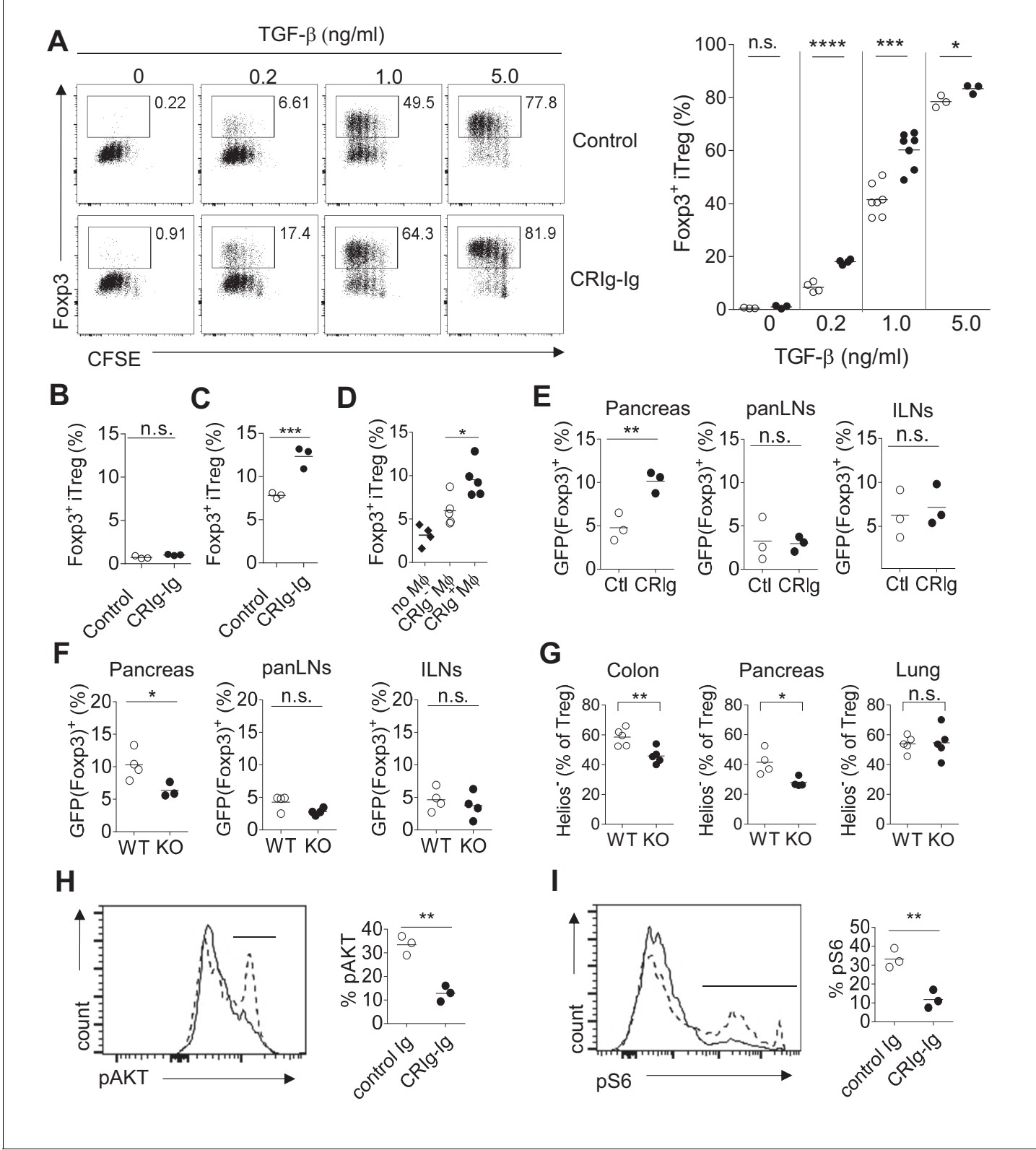

**Figure 3.** CRIg promotes iTreg generation in vitro. (A) (left) Representative FACS plots depicting the generation of iTreg cells in the presence of CRIg-Ig, or control Ig, under various concentrations of TGF-β. (right) Statistics of multiple experiments. (B) Tconv cells were cultured in the condition of anti-CD3/CD28 and IL-2, anti-TGF-β neutralizing antibody (clone 1D11) and either control Ig, or CRIg-Ig. (C) Total splenocytes from 8-week-old NOD/BDC2.5/Thy1.1 mice were labeled with CTV and cultured with BDC2.5 mimotope (100 ng/ml) for 3 days. The generation of Treg cells were analyzed by

*Figure 3 continued on next page*

*Figure 3 continued*

intracellular Foxp3 staining. (D) iTreg generation in vitro as in (C) with the inclusion of CRIg$^+$ or CRIg$^-$ macrophages, sorted from peritoneal cavity. (E) Purified CD4$^+$Foxp3 (GFP)$^-$ T cells from NOD/BDC2.5/Foxp3$^{GFP}$/Thy1.1 mice were transferred into 4-week-old NOD mice, followed by *i.p.* injection of CRIg-Ig, or control Ig every other day for 2 weeks. (F) Purified CD4$^+$Foxp3 (GFP)$^-$ T cells from NOD/BDC2.5/Foxp3$^{GFP}$/Th1.1 mice were transferred into 4-week-old NOD or NOD/CRIg KO mice. The generation of Foxp3(GFP)$^+$ cells in pancreatic islets, panLNs and inguinal LNs(ILNs) was analyzed 2 weeks after the transfer. (G) Flow cytometric analyses of Helios$^-$ Treg cells from pancreas, colon and lung of NOD/CRIgKO and wildtype controls. (H, I) Purified CD4$^+$ Foxp3(GFP)$^-$ T cells were cultured with either control Ig or CRIg-Ig for 18 hr and analyzed for the phosphorylation of AKT (H) and ribosomal protein S6 (I). Dotted line, control Ig; solid line, CRIg-Ig. Data are representative of five (A), three (B–I) experiments. Student's *t*-test was used. n.s., non-significant. *p<0.05; ***p<0.001; ****p<0.0001. Ctl, control; KO, CRIg knockout.

DOI: https://doi.org/10.7554/eLife.29540.011

The following figure supplements are available for figure 3:

**Figure supplement 1.** CRIg-Ig does not enhance TGF-β induced phosphorylation of Smad2/3.
DOI: https://doi.org/10.7554/eLife.29540.012
**Figure supplement 2.** Experimental design and FACS profiles of in vivo pTreg generation promoted by CRIg-Ig.
DOI: https://doi.org/10.7554/eLife.29540.013

generation of iTreg cells (*Figure 3A*). CRIg-mediated enhancement of iTreg generation was more profound when TGF-β concentration was relatively low (e.g., there was a more than two-fold increase of iTreg induction by CRIg-Ig with 0.2 ng/ml TGF-β) (*Figure 3A*). To assess whether CRIg-Ig alone could induce the conversion of Tconv cells into iTreg cells, an anti-TGF-β neutralizing mAb (clone 1D11) (ref. [*Chang et al., 2015*]) was added to eliminate any trace-amount of TGF-β in the culture medium. In the absence of TGF-β, CRIg-Ig alone did not induce the generation of iTreg cells (*Figure 3B*). Treg cells can be converted from Tconv cells under the conditions where antigens (Ag) are encountered in the absence of optimal costimulation (*Kretschmer et al., 2005*; *Verginis et al., 2008*). To examine whether CRIg has an effect on Ag-induced Treg generation, we cultured total splenocytes from NOD/BDC2.5 mice with the cognate Ag mimotopes (*Haskins et al., 1988*; *Katz et al., 1993*). Compared with control Ig, CRIg-Ig induced a 2-fold increase of Treg generation (*Figure 3C*), suggesting a role of CRIg in promoting Ag-stimulated Treg generation. We next evaluated whether CRIg$^+$ macrophages can promote iTreg generation. Using an in vitro Ag-stimulated iTreg generation condition, we found that even though CRIg$^-$ macrophages can modestly promote Ag-induced iTreg generation, this effect was significantly elevated by CRIg$^+$ macrophages (*Figure 3D*). Therefore, CRIg promotes iTreg generation induced by either TGF-β, or Ag.

To evaluate the in vivo effect of CRIg in peripherally Treg (pTreg) generation, we transferred highly purified CD4$^+$ Foxp3(GFP)$^-$ Tconv cells isolated from NOD/BDC2.5/Thy1.1$^+$/Foxp3$^{GFP}$ mice into 4-week-old NOD mice. We found that a 2-week in vivo CRIg-Ig treatment preferentially enhanced pTreg development in pancreatic islets, not pancreas- draining or non-draining lymph nodes (*Figure 3E*). Conversely, we transferred purified BDC2.5$^+$ CD4$^+$ Foxp3(GFP)$^-$ Tconv cells into either NOD or NOD/CRIg KO mice. The generation of Treg cells was compromised in the absence of CRIg (*Figure 3F*). Of note, this reduction only occurred in the pancreas, not in pancreas-draining or non-draining lymph nodes, suggesting a local microenvironment-specific role of CRIg in regulating pTreg generation. Consistent with the role of CRIg in affecting Treg cells at tissue sites, we found that Helios$^-$ peripherally-induced Treg (pTreg) (*Shevach and Thornton, 2014*) cells were preferentially reduced in pancreas and colon of NOD/CRIg KO mice, compared with wildtype controls (*Figure 3G*). In contrast, since there was no CRIg expression in lung macrophages (*Figure 1—figure supplement 1*), pTreg cells in the lung were not affected by CRIg deficiency.

We next investigated what pathways were affected in CRIg-promoted iTreg cells. It has been demonstrated that mTOR pathway kinase AKT has a negative impact on Treg cell development (*Haxhinasto et al., 2008*). We asked whether CRIg affected AKT activity during iTreg differentiation. Under iTreg induction condition, plate-bound CRIg-Ig more potently suppressed the phosphorylation of AKT (*Figure 3H*). Similarly, the phosphorylation of S6 ribosomal protein, a downstream target of mTOR pathway was also more significantly suppressed by CRIg-Ig (*Figure 3I*). However, CRIg-Ig did not enhance TGF-β induced phosphorylation of Smad2/3 (*Figure 3—figure supplement 1*). Thus, CRIg-Ig potentiates iTreg generation via suppressing AKT-MTOR pathway.

## CRIg stabilizes Foxp3 expression in iTreg cells

iTreg cells lose their expression of Foxp3 when they are restimulated with TCR signaling (*Feng et al., 2014*; *Yue et al., 2016*). We asked whether CRIg-Ig affected the stability of iTreg cells. CD4$^+$ Foxp3(GFP)$^-$ cells from NOD/Foxp3$^{GFP}$ mice were FACS purified and cultured under the iTreg differentiation condition. After 24 hr, about 20% of cultured cells were Foxp3(GFP)$^+$. We sorted these GFP$^+$ cells and recultured them with anti-CD3/CD28 and IL-2, in the presence of either CRIg-Ig, or control Ig (*Figure 4A*). An anti-TGF-β neutralizing antibody (clone 1D11) was added to eliminate the trace amount of TGF-β in the culture medium. After 3 days, about 50% of control iTreg cells lost their expression of Foxp3 (*Figure 4B*). In contrast, in the presence of plate-bound CRIg-Ig, a significantly higher fraction of iTreg cells retained Foxp3 (*Figure 4B*). The loss of Foxp3 in restimulated iTreg cells was exacerbated over the course of cell dividing. This instability of Foxp3 expression was repressed by CRIg-Ig, especially when the cells underwent multiple divisions (*Figure 4C*). Moreover, the expression level of Foxp3 protein on a per cell basis was also higher in restimulated iTreg cells in the presence of CRIg-Ig, compared to that in control group (*Figure 4D*). Notably, this Treg-stabilizing effect of CRIg is TGF-β independent, because we have added an anti-TGF-β neutralizing mAb to eliminate any trace amount of TGF-β. Thus, CRIg retains the expression of Foxp3 in Treg cells. In line with a Treg–stabilizing effect, CRIg preconditioned iTreg cells exhibited an enhanced suppressive capacity in an in vitro Treg suppression assay (*Figure 4—figure supplement 1*).

## CRIg-Ig stabilizes iTreg cells by enhancing their responsiveness to IL-2

We next attempted to identify the mechanisms by which CRIg stabilized Foxp3 in Treg cells. Demethylation of CpG sites in the second CNS region (CNS2) of *Foxp3* is critical for Treg stability (*Floess et al., 2007*; *Zheng et al., 2010*). We asked whether CRIg-Ig had an effect on *Foxp3* CpG demethylation. We used bisulfite colony sequencing of PCR products of CNS2 regions (*Kalekar et al., 2016*). To this end, iTreg cells were generated, sorted as GFP(Foxp3)$^+$ cells and recultured with anti-CD3/CD28 and IL-2, in the presence of CRIg-Ig, or control Ig. After 3 days, genomic DNAs from re-sorted GFP$^+$ cells were extracted and processed for bisulfite sequencing of the *Foxp3* CNS2 region. As expected, CpG sites in *Foxp3* CNS2 region of control iTreg cells were highly methylated (*Figure 4E*). A similar profile of CpG methylation was observed in CRIg iTreg cells (*Figure 4E*). These data suggest that CRIg-promoted iTreg stability is not a consequence of demethylation in *Foxp3* CNS2 region.

IL-2 signaling is critical for Treg stability, by retaining the expression of Foxp3 (refs [*Dépis et al., 2016*; *Chen et al., 2011*]). We asked whether iTreg cells, when restimulated in the presence of CRIg-Ig, were more responsive to limited amount of IL-2. In this regard, TGF-β induced iTreg cells were sorted and restimulated with anti-CD3/CD28, in the presence of CRIg-Ig or control Ig, with various doses of IL-2. In control iTreg cells restimulated with anti-CD3/CD28, increased concentrations of IL-2 did not prevent the loss of Foxp3 in these cells. In contrast, the presence of CRIg-Ig resulted in a significantly higher fraction of restimulated iTreg cells retaining their expression of Foxp3, in the presence of IL-2 (*Figure 4F*). CRIg induced iTreg cells expressed a higher level of IL-2Rβ (*Figure 4G*), suggesting an enhanced responsiveness of these cells to IL-2. Signal transducer and activator of transcription 5 (STAT5) is key downstream target of IL-2 signaling (*Burchill et al., 2007*; *Yu et al., 2009*). IL-2-induced STAT5 phosphorylation in iTreg cells was significantly enhanced by CRIg-Ig (*Figure 4H*). Therefore, CRIg retains Foxp3 expression in iTreg cells by enhancing their responsiveness to IL-2.

## CRIg stabilizes Treg cells in vivo in lymphopenic condition

Treg cells lose Foxp3 expression when they are transferred into lymphopenic mice (*Duarte et al., 2009*; *Bailey-Bucktrout and Bluestone, 2011*). Using neonatal NOD mice as lymphopenic hosts, we analyzed whether CRIg-preconditioned iTreg cells were more stable in vivo. CD4$^+$ Foxp3(GFP)$^-$ cells from NOD/Foxp3$^{GFP}$/Thy1.1$^+$ mice, or NOD/Foxp3$^{GFP}$/Thy1.1$^+$ Thy1.2$^+$ mice were sorted and differentiated into iTreg cells by either TGF-β (control iTreg), or TGF-β plus CRIg-Ig (CRIg iTreg). Both control iTreg cells and CRIg iTreg cells were sorted as GFP$^+$ cells and mixed at 1:1 ratio before transferring into neonatal NOD mice (Thy1.2$^+$) (*Figure 5A*). One week later, transferred iTreg cells from both spleen and pancreatic islets were analyzed. Control and CRIg iTreg cells exhibited similar degrees of engraftments (*Figure 5B,C*), suggesting that preconditioning with CRIg did not affect

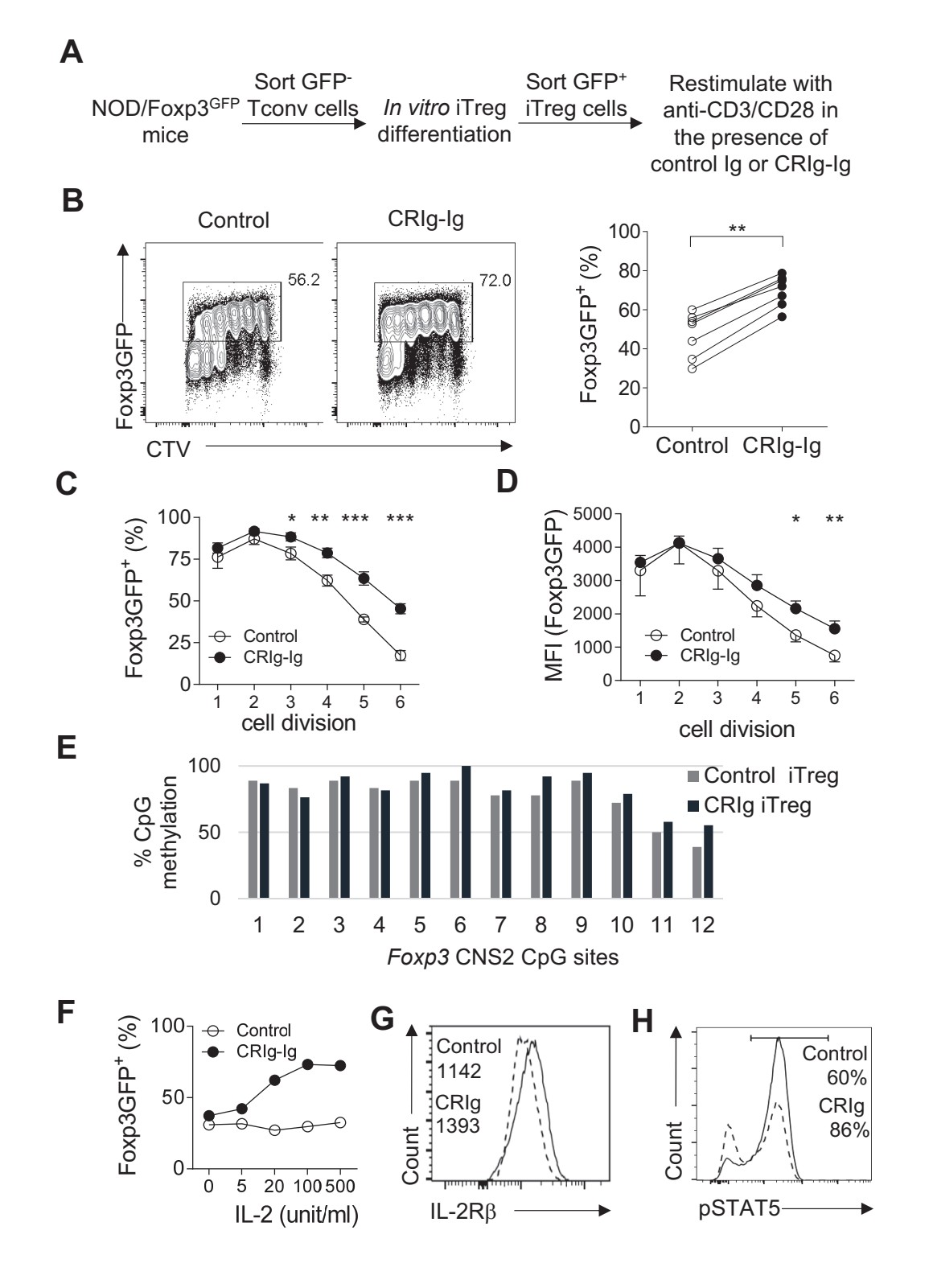

**Figure 4.** CRIg stabilizes Foxp3 expression in TGF−β induced iTreg cells. (**A**) Experimental setting. (**B**) The fraction of Foxp3 positivity and the division of recultured iTreg cells. Left, representative FACS plots; right, the statistics of multiple experiments (n = 7). (**C**) The percentage of cells retaining Foxp3 expression in each cell division. (**D**) The MFI of Foxp3 protein in each generation of cell division. (**E**) The methylation percentage at each CpG motif in *Foxp3* CNS2 of control iTreg (grey bars), or CRIg iTreg cells (black bars) (see *Figure 4—source data 1*) (**F**) In vitro differentiated iTreg cells were
*Figure 4 continued on next page*

*Figure 4 continued*

restimulated with anti-CD3/CD28, and various concentrations of IL-2, in the presence of CRIg-Ig, or control Ig. The fraction of cells retaining Foxp3 expression was analyzed after 3 days. (G) The expression of IL-2Rβ in control and CRIg iTreg cells after 3 days of culture. (H) The phosphorylation of STAT5 in control and CRIg iTreg cells. Data are representative of seven (**B–D**), two (**E**), and three (**F–H**) experiments, respectively. Student's t-test was used. *p<0.05; **p<0.01; ***p<0.001.

DOI: https://doi.org/10.7554/eLife.29540.014

The following source data and figure supplement are available for figure 4:

**Source data 1.** The methylation percentage at each CpG motif in *Foxp3* CNS2 of control iTreg cells, CRIg iTreg cells and ex vivo Treg cells (associated with **Figure 4E**).
DOI: https://doi.org/10.7554/eLife.29540.016
**Figure supplement 1.** CRIg enhances iTreg suppressive function.
DOI: https://doi.org/10.7554/eLife.29540.015

homeostatic survival and expansion of these adoptively transferred cells. However, iTreg cells generated in the presence of CRIg-Ig showed an enhanced stability under lymphopenic condition. In spleen, a significantly higher fraction of transferred CRIg-iTreg cells maintained their expression of Foxp3 (*Figure 5B*). Although CRIg (plate-bound) was present during the in vitro generation of iTreg cells, it was dissociated from these cells after adoptive transfer. Since there was no CRIg expression in spleen (*Helmy et al., 2006*; *Vogt et al., 2006*) (and our unpublished data), it thus suggests that a continuous cell-cell contact between Treg cells and CRIg⁺ TRMs is not necessarily needed for retaining Foxp3 expression. The instability of adoptively transferred iTreg cells was more profoundly exacerbated in pancreatic islets, as expected. However, preconditioning with CRIg before the adoptive transfer retained a significantly higher percentage of Foxp3 expression (*Figure 5C*). Thus, iTreg cells generated by TGF-β plus CRIg-Ig are more stable in vivo.

## In vivo CRIg modulation corrects immune dysregulation in an autoimmune condition

Immune dysregulation characterized by loss-of balance between regulatory and effector immune cells is key for the pathogenesis of many autoimmune diseases, including T1D (*Bluestone et al., 2010*; *Anderson and Bluestone, 2005*). Our data suggested that CRIg may possess a role in modulating the dysregulated immune responses. We assessed this possibility in NOD mice, the primary animal models for human T1D (*Anderson and Bluestone, 2005*). Our in vitro studies revealed that cross-linking CRIg with an anti-CRIg mAb (e.g., clone 17C9) significantly enhanced TGF-β-induced iTreg generation (*Figure 6—figure supplement 1*). In vivo analyses revealed that serum half-life of CRIg-Ig was significantly prolonged when an anti-CRIg mAb (clone 17C9) was administrated simultaneously with CRIg-Ig (*Figure 6—figure supplement 2*). Moreover, anti-CRIg mAb alone did not alter the compositions of T cells in vivo, including Treg cells (*Figure 6—figure supplement 3A*). Therefore, we used CRIg-Ig/anti-CRIg mAb complex to treat NOD mice. Ten-week old pre-diabetic NOD mice were treated with CRIg-Ig/anti-CRIg complex or control Ig for two weeks. Immune cells isolated from the spleen, pancreatic draining lymph nodes (panLNs) and pancreatic islets were analyzed. We first focused on analyzing whether in vivo CRIg modulation altered the balance between regulatory and effector T cells in NOD mice. Of note, the ratios between Treg cells and CD4⁺ Tconv cells in pancreatic islets were significantly increased in CRIg-Ig/anti-CRIg complex treated mice, with about 50% of the mice exhibited a robust response (*Figure 6A*). Similar results were observed in the ratios between Treg and CD8⁺ T cells (*Figure 6A*). Of note, such an effect was only seen at tissue sites (pancreatic islets), not in lymphoid organs, such as panLNs (*Figure 6B*), or spleen (*Figure 6—figure supplement 3B*). Thus, in vivo CRIg modulation rebalances immune compositions in situ under an autoimmune condition.

In addition to altering the ratios between regulatory and effector T cells, we asked whether CRIg affected the functional properties of T cells. Interestingly, in vivo CRIg modulation increased the proportion of Helios⁻ Treg cells (*Figure 6C*), indicating an enhanced generation of pTreg cells (*Thornton et al., 2010*). This is consistent with the results of CRIg-Ig promoted in vivo pTreg generation from adoptively transferred Tconv cells (*Figure 3E*). Elevated expression of ICOS is a marker for effector Treg cells (*Smigiel et al., 2014*). We found that a significantly increased proportion of islet

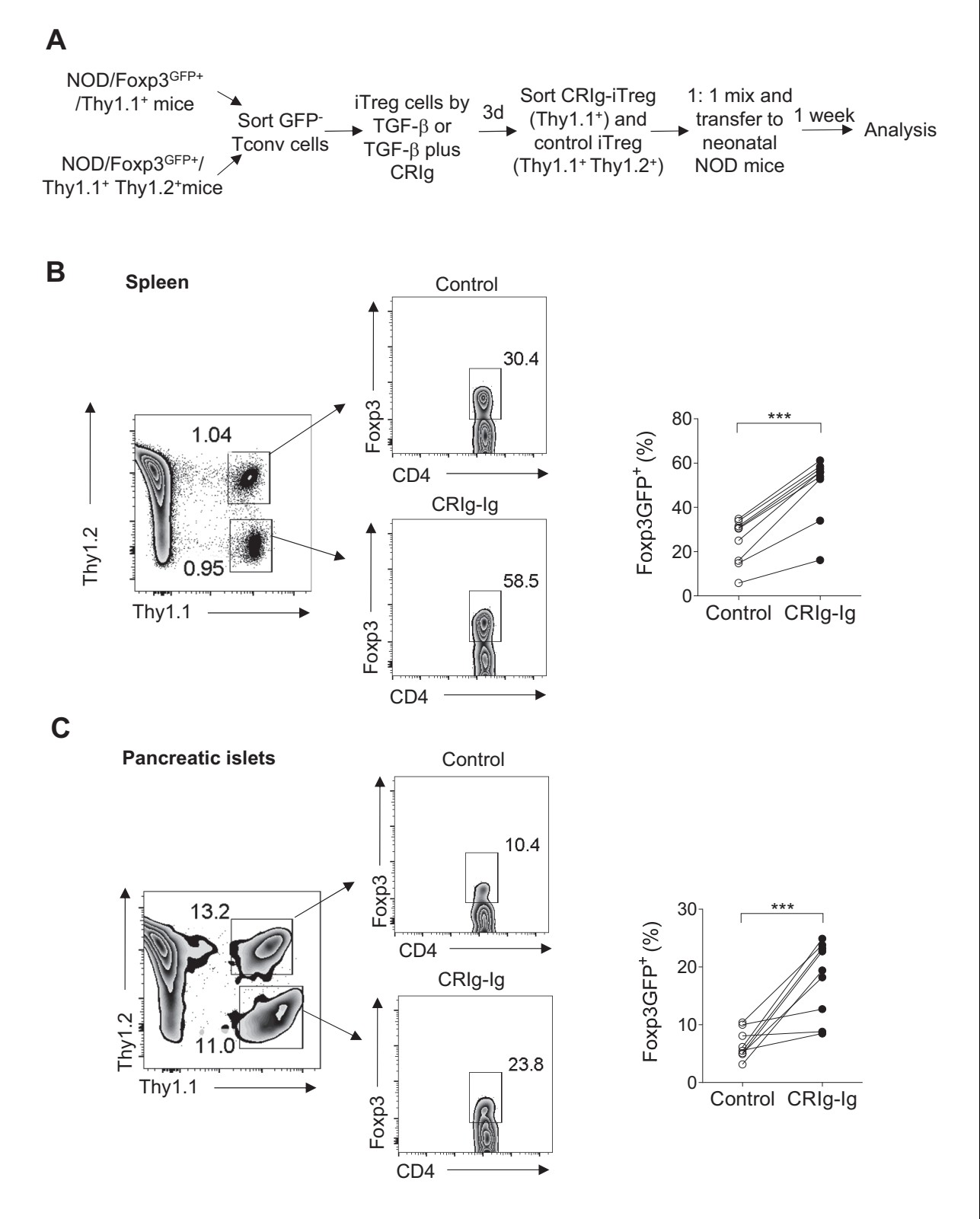

**Figure 5.** CRIg stabilizes adoptively transferred iTreg cells in vivo. (A) Experimental design. (B, C) One week later, the transferred cells were isolated from spleen (B) and pancreatic islets (C) and were analyzed for the expression of Foxp3. Control iTreg cells, Thy1.1[+] Thy1.2[+]; CRIg iTreg cells, Thy1.1[+] Thy1.2[-]. Data are representative of two independent experiments with 9 mice in total. Student's t-test was used. ***p<0.001.
DOI: https://doi.org/10.7554/eLife.29540.017

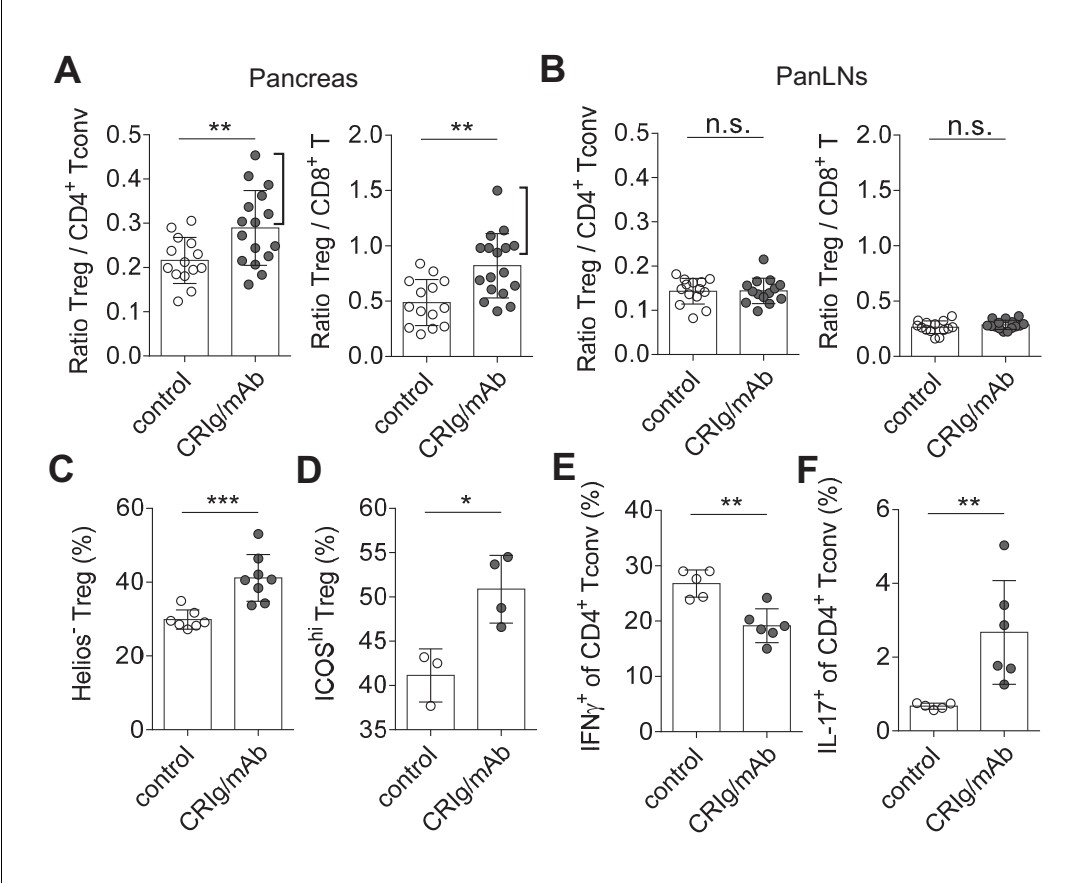

**Figure 6.** CRIg restores immune tolerance in pancreatic islets of NOD mice. The ratios of Treg/Tconv cells, and Treg/CD8[+] T cells in the pancreas (**A**) and panLNs (**B**) of control (n = 14) and CRIg-Ig/anti-CRIg treated mice (n = 16). The expression of Helios (**C**) and ICOS (**D**) in Treg cells from pancreatic islets of control (n = 7 in C, n=3 in D) and CRIg-Ig/anti-CRIg (labeled as CRIg/mAb in the figure panels, n = 8 in C, n=4 in D) treated mice. The production of IFN-γ (**E**) and IL-17 (**F**) in CD4[+] Tconv cells from pancreatic islets in control Ig (n = 5) and CRIg-Ig/anti-CRIg (n = 6) treated mice. Data are representative of six (**A, B**), three (**C**), and two (**D–F**) experiments, respectively. Student's t-test was used. n.s., non-significant; *p<0.05; **p<0.01; ***p<0.001.

DOI: https://doi.org/10.7554/eLife.29540.018

The following figure supplements are available for figure 6:

**Figure supplement 1.** Cross-linking CRIg with anti-CRIg mAb enhances iTreg generation.

DOI: https://doi.org/10.7554/eLife.29540.019

**Figure supplement 2.** Anti-CRIg mAb prolongs in vivo half-life of CRIg-Ig.

DOI: https://doi.org/10.7554/eLife.29540.020

**Figure supplement 3.** In vivo modulation of CRIg in NOD mice.

DOI: https://doi.org/10.7554/eLife.29540.021

Treg cells from CRIg-Ig/anti-CRIg treated mice were ICOS[hi] (*Figure 6D*). Thus, in vivo CRIg modulation increases pTreg development and functional maturation at tissue sites.

As a consequence of CRIg on helper T (Th) cell differentiation, we observed a significantly reduced IFN-γ production in CD4[+] Tconv cells from pancreatic islets (*Figure 6E*). However unexpectedly, IL-17[+] T (Th17) cell differentiation was significantly increased in pancreatic islets of CRIg-Ig/anti-CRIg complex treated mice (*Figure 6F*). The role for Th17 cells in T1D has been controversial, either diabetogenic (*Emamaullee et al., 2009*; *Gao et al., 2010*) or protective (*Tse et al., 2010*; *Nikoopour et al., 2010*). One model of Th17's role in T1D is that these cells convert to diabetogenic IFN-γ[+] Th1 cells in vivo (*Martin-Orozco et al., 2009*; *Bending et al., 2009*). Because we have previously found that the onset of diabetes was suppressed in CRIg-Ig treated NOD mice (*Fu et al., 2012*), it thus can be speculated that these accumulated Th17 cells in pancreatic islets are not

diabetogenic. The elevated proportion of IL-17+ T cells in pancreatic islets of CRIg-treated NOD mice may be due to a blocked conversion from Th17 into Th1 cells.

Thus, in vivo CRIg modulation corrects immune dysregulation occurred in autoimmune conditions. Such an outcome is achieved through a dual effect of CRIg: the suppression of pathogenic Th1 cells and the promotion of Treg cell differentiation and function. The diabetogenicity of effector T cells can be further dampened by elevated Treg abundance and function. Overall, CRIg-mediated immunomodulation diverts autoimmunity towards immunological tolerance.

## The expression of CRIg in TRMs is influenced postnatally by microbial signals

The above findings support a role of CRIg as a tissue environment-specific checkpoint molecule for immunological homeostasis. The dearth of CRIg expression in TRMs is tightly correlated with exacerbated tissue inflammation (*Figure 1*). Therefore, it is worth investigating what factors and signals regulate the expression of CRIg in TRMs. CRIg is abundantly expressed in a number of organs/tissues, including liver Kupffer cells of adult mice (*Figure 7A*) (*Helmy et al., 2006*). However, in contrast, CRIg was almost completely absent in the liver from neonatal (day one post birth) mice, though F4/80+ Kupffer cells were readily detected (*Figure 7A*). We confirmed the absence of CRIg expression in macrophages from other tissues, including the pancreas in neonatal mice (*Figure 7B*). Therefore, CRIg is not constitutively expressed in macrophages, rather, is contingent upon extrinsic signals arisen postnatally. We then asked when the expression of CRIg was induced postnatally. For this, we tracked the expression of CRIg in peritoneal large TRMs (F4/80hi) longitudinally in both NOD and B6 strains (*Figure 7C* and *Figure 7—figure supplement 1A*). Clearly, there was a sharp increase of CRIg expression in macrophages around the time of weaning, a time-point when mice were exposed to environmental cues (*Laukens et al., 2016*). The colonization of gut microbiota at early age, especially around the time of weaning is critical for the postnatal development of immune system. To evaluate whether the expression of CRIg was influenced by gut microbiota, we treated adult B6 mice with a combination of four antibiotics (vancomycin, metronidazole, neomycin and ampicillin, *VMNA*) (*Sefik et al., 2015*) for one week in drinking water. We analyzed the expression of CRIg in various tissues, including pancreas, colon and peritoneal cavity. Though the fraction of CRIg+ TRMs in peritoneal cavity was not affected by antibiotic treatment, in contrast, at tissue sites (colon and pancreas), the fractions of CRIg+ TRMs were significantly decreased (*Figure 7D–F*). Of note, at all examined locations, antibiotic treatment significantly diminished CRIg expression on a per cell basis in macrophages (*Figure 7D–F*). These data suggest that CRIg expression in TRMs is induced postnatally by microbial signals.

## Retinoic acid signaling retains CRIg expression in macrophages in vitro and in vivo.

We have found that TRMs (both CRIg+ and CRIg- subsets) abundantly express retinoic acid (RA) receptors (RAR) (*Figure 7G*), suggesting that these cells are able to respond to RA. Therefore, we tested whether RA affected CRIg expression. We isolated CRIg+ macrophages from peritoneal cavity of 7-week-old B6 mice and cultured them in the presence, or absence of *all-trans* RA (ATRA), the active form of RA. In the absence of ATRA, the expression of CRIg decreased (reflected by a lower MFI of CRIg) after 3-day culture. In contrast, ATRA retained the expression of CRIg, to a level comparable to that in freshly isolated CRIg+ macrophages (*Figure 7H*). Interestingly, ATRA also increased CRIg expression in isolated CRIg- macrophages (*Figure 7—figure supplement 1B*), suggesting that RA signaling may be involved in inducing CRIg expression in macrophages. *Might RA affect CRIg expression* in vivo? To test this, we treated 7-week-old B6 mice with an inverse pan-RA agonist BMS 493 (ref [*Chazaud et al., 2003*]) to block RA signaling. Compared with control vehicle-treated mice, BMS 493 significantly reduced CRIg expression in peritoneal, but not pancreatic F4/80hi TRMs (*Figure 7I,J*). Consistently, BMS 493 treatment also reduced the expression of CRIg in peritoneal macrophages in an adoptive transfer setting, where donor CRIg+ macrophages were purified from congenically-labeled mice (*Figure 7—figure supplement 1C*). Collectively, these data suggest that RA signaling is involved in regulating CRIg expression in TRMs in a tissue-specific manner.

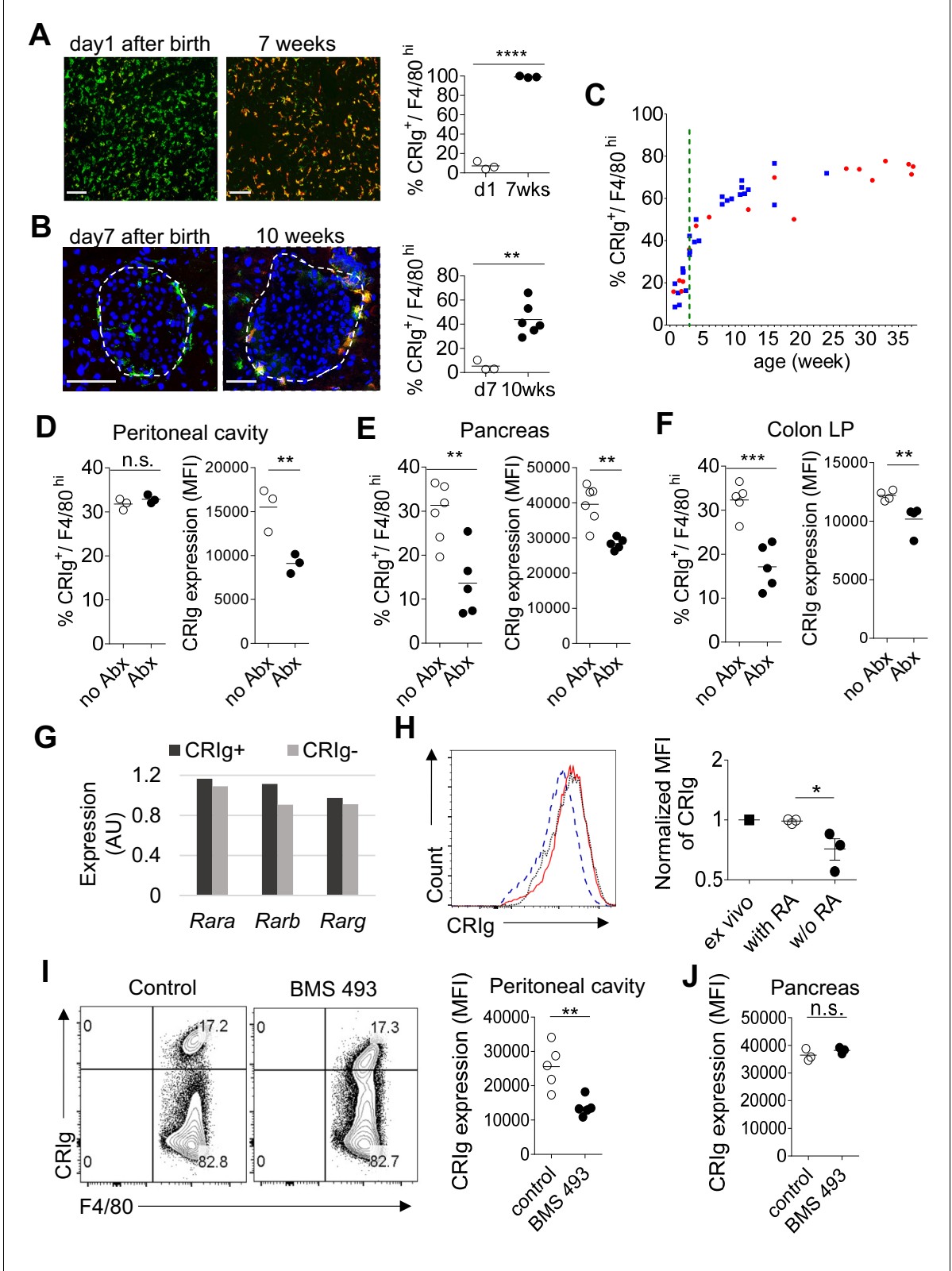

**Figure 7.** The expression of CRIg in TRMs is influenced by environmental factors. (**A**) The expression of CRIg in liver Kupffer cells of adult (7 weeks of age) and neonatal (day one post birth) B6 mice. Green, F4/80; Red, CRIg. (**B**) The expression of CRIg in pancreatic islets of adult (10 weeks of age) and neonatal (day seven post birth) NOD mice. Green, F4/80; Red, CRIg. (**C**) Longitudinal analysis of CRIg expression in peritoneal F4/80hi TRMs of NOD mice. Red: females; Blue, males. Dotted green line depicts the time of weaning. (**D–F**) The percentages and the MFI of CRIg

*Figure 7 continued on next page*

*Figure 7 continued*

expression in TRMs from peritoneal cavity (D), pancreatic islets (E) and colon (F) of control and antibiotics treated B6 mice, respectively. (G) The expression of retinoic acid receptors in CRIg$^+$ and CRIg$^-$ TRMs isolated from peritoneal cavity of 7 weeks old B6 mice. (H) CRIg$^+$ macrophages were isolated from peritoneal cavity and cultured in vitro in the presence or absence of ATRA. The expression of CRIg in cultured cells were analyzed 3 days later. blue line, no ATRA; red line, with ATRA. (I, J) In vivo treatment of 7 weeks old B6 mice with an inverse pan-RA agonist BMS 493 to block RA signaling. The expression of CRIg in TRMs from peritoneal cavity (I) and pancreatic islets (J) was analyzed 3 days later. Data are representative of three (A, B, D–J), and more than five (C) experiments, respectively. Student's t-test was used. n.s., non-significant; *p<0.05; **p<0.01; ***p<0.001; ****p<0.0001. w/o, without.

DOI: https://doi.org/10.7554/eLife.29540.022

The following figure supplement is available for figure 7:

**Figure supplement 1.** RA signaling is involved in CRIg expression in TRMs.
DOI: https://doi.org/10.7554/eLife.29540.023

## Discussion

How adaptive immunity at tissue sites is regulated by local microenvironment remains poorly understood. We here define a new model of regulating T cell activities by tissue resident cells. In this model, TRMs sense environmental cues and express CRIg. Through the function of CRIg, tissue-specific autoimmunity is suppressed and immune tolerance is reinforced. Multiple mechanisms are involved in this model to promote immune tolerance. First, CRIg$^+$ TRMs form a barrier protecting pancreatic islets from succumbing to destructive insulitis. Second, CRIg preferentially suppresses effector T cell proliferation. Third, CRIg promotes the differentiation and stability of Treg cells.

CRIg$^+$ TRM mediated protection provides one explanation why not all islets are equally destructed during T1D development. In animal models and arguably in human patients of T1D, islet autoimmunity (insulitis) is universally present. However, not all insulitis proceeds to become diabetic (*Fu et al., 2012*; *André et al., 1996*; *Poirot et al., 2004*). Even in genetically identical monozygotic twins, the concordance of T1D incidence is only about 40–50%. It is still elusive what determines the risk of one individual to become diabetic. Moreover, not all islets are equally destroyed and there exist intact islets even in longstanding diabetic patients (*Bluestone et al., 2010*). It is unclear why some islets are free from immune infiltration while the others are not. CRIg$^+$ TRMs specifically localize in the capsular area of an islet. The dearth of CRIg expression in these TRMs is correlated with the aggressiveness of insulitis in each individual islet. In line with this, insulitis is markedly exacerbated in NOD mice with CRIg-null mutation.

Several aspects of T cell activities can be modulated by CRIg. For each of them, our data also provide critical mechanistic insights. First, CRIg suppresses T cell proliferation by attenuating the activation of multiple TCR signaling cascades, including the most proximal signaling complex, the ERK-MAPK pathway and the AKT-MTOR pathway. The preferential suppression of effector T cells by CRIg is likely due to a higher level of the expression of a putative CRIg receptor in effector T cells than that in Treg cells. Our binding assays suggest that the receptor for CRIg is only transiently induced during early activation phase in T cells. Using competition assays, we have excluded a number of known coinhibitory molecules of immunoglobulin family as a potential receptor for CRIg. Other approaches including biochemical assays and protein interaction arrays will be employed to identify the receptor for CRIg. Second, reminiscent of the role of PD-L1 (refs [*Francisco et al., 2009*; *Wang et al., 2008*]), CRIg can also promote TGF-β induced Treg differentiation. CRIg does not directly enhance TGF-β signaling, instead, it robustly suppresses AKT signaling under iTreg induction condition, thus potentiating the differentiation of iTreg cells. Beyond promoting iTreg differentiation, another key finding is that CRIg increases the stability of Treg cells. This Treg-stabilizing effect of CRIg is not attributed to epigenetic modification of *Foxp3* promoter, instead, CRIg makes Treg cells more sensitive to IL-2 by increasing the expression of IL-2Rβ and the phosphorylation of STAT5. Moreover, our data suggest that Treg cells do not need continuous cell-cell contact with CRIg$^+$ TRMs to maintain Foxp3 expression.

Adoptive transfer of Treg cells is a promising strategy to induce immune tolerance in the treatment of human autoimmune diseases and organ transplantations (*Tang and Bluestone, 2013*). However, variable stabilities of in vitro generated iTreg cells hinder this therapeutic option. We propose that CRIg may become a potential candidate to overcome the barrier of instability in Treg-based

therapies. Correlated with the enhanced stability of CRIg iTreg cells, these cells are also more suppressive in a T cell proliferation assay.

CRIg has also been reported as a complement receptor (*Helmy et al., 2006*). However, our data reveal that the effects of CRIg in T cells can be exerted in a complement-independent manner. This suggests that complement components are not necessarily involved in CRIg-mediated suppression of effector T cells and differentiation of Treg cells.

CRIg emerges as a molecular link between environmental cues and adaptive immunity in controlling excessive tissue inflammation. The expression of CRIg in macrophages is dampened by inflammatory signals, such as lipopolysaccharide (*Vogt et al., 2006*). Loss of abundance of CRIg$^+$ macrophages is associated with exacerbated tissue inflammation in multiple autoimmune conditions (*Vogt et al., 2006*; *Fu et al., 2012*). Therefore, identifying what signals induce or suppress the expression of CRIg in TRMs is essential for better understanding how TRMs sense environmental signals and in turn orchestrate the activities of T cells. We find that CRIg expression in TRMs is induced postnatally and the abundance of CRIg$^+$ TRMs sharply increases around weaning, a time window allowing postnatal colonization of gut microbiota in mice (*Laukens et al., 2016*). The connection between gut microbiome and CRIg is evident from our studies showing that antibiotic treatment diminishes CRIg expression in TRMs at various locations, including colon, pancreas and peritoneal cavity. A recent study has demonstrated that antibiotic-mediated gut dysbiosis accelerated the development of T1D in mice (*Livanos et al., 2016*). The dampened expression of CRIg in pancreatic TRMs under dysbiotic conditions could be one plausible mechanism. Thus, environmental factors (most likely microbial signals and metabolites) instruct TRMs to express CRIg, which functions as an immune checkpoint molecule to regulate tissue T cell activities.

We found CRIg$^+$ cells in human pancreas, suggesting a potential role of CRIg in human T1D. CRIg$^+$ TRMs are also abundantly present in other gastrointestinal organs, including liver and intestines in both mice and humans (*Helmy et al., 2006*). It thus can be speculated that CRIg may also play a role in maintaining immunological homeostasis in these tissues.

## Materials and methods

**Key resources table**

| Reagent type (species) or resources | Designation | Source or reference | Identifiers | Additional information |
| --- | --- | --- | --- | --- |
| strain, strain background (*Mus musculus*) | C57BL/6 (B6) mice | The Jackson Laboratory | RRID: IMSR_JAX:000664 | |
| strain, strain background (*Mus musculus*) | NOD mice | Mathis-Benoist laboratory | RRID: IMSR_JAX:001976 | |
| strain, strain background (*Mus musculus*) | NOD/Foxp3$^{GFP}$ mice | The Jackson Laboratory | RRID: IMSR_JAX:025097 | |
| strain, strain background (*Mus musculus*) | B6/CD45.1 mice | Dr. Li-Fan Lu, UCSD | RRID: IMSR_JAX:002014 | |
| strain, strain background (*Mus musculus*) | NOD/BDC2.5 /Thy1.1 mice | Mathis-Benoist laboratory | | |
| strain, strain background (*Mus musculus*) | NOD/BDC2.5 /Foxp3$^{GFP}$ /Thy1.1 mice | this paper | | NOD/BDC2.5/Foxp3$^{GFP}$/ Thy1.1 mice were generated by crossing NOD/BDC2.5/Thy1.1 mice to NOD/ Foxp3$^{GFP}$ mice. |

*Continued on next page*

*Continued*

| Reagent type (species) or resources | Designation | Source or reference | Identifiers | Additional information |
|---|---|---|---|---|
| strain, strain background (*Mus musculus*) | NOD/BDC2.5 /Foxp3$^{GFP}$ /Thy1.1/Thy1.2 mice | this paper | | NOD/BDC2.5/Foxp3$^{GFP}$/ Thy1.1/Thy1.2 mice were generated by crossing NOD/BDC2.5/Thy1.1 mice to NOD/Foxp3$^{GFP}$ mice. |
| strain, strain background (*Mus musculus*) | B6/CRIg KO mice | Genentech | | PMID: 16530040 |
| strain, strain background (*Mus musculus*) | NOD/CRIg KO mice | this paper | | NOD/CRIg$^{-/-}$ mice were generated by crossing B6/CRIg$^{-/-}$ mice onto a NOD background for more than 10 generations. |
| biological sample (*Mus musculus*) | pancreas | other | | prepared from NOD mice |
| biological sample (*Mus musculus*) | colon lamina propria | other | | prepared from NOD orB6 mice |
| biological sample (*Mus musculus*) | peritoneal cavity cells | other | | prepared from NOD or B6 mice |
| biological sample (*Mus musculus*) | lung | other | | prepared from NOD or B6 mice |
| biological sample (*Mus musculus*) | liver | other | | prepared from NOD or B6 mice |
| biological sample (*Mus musculus*) | serum | other | | prepared from NOD mice |
| biological sample (*Mus musculus*) | panLNs | other | | pancreatic draining lymph nodes from NOD mice |
| biological sample (*Mus musculus*) | ILNs | other | | Inguinal lymph nodes from NOD or B6 mice |
| biological sample (*Mus musculus*) | spleen | other | | prepared from NOD or B6 mice |
| biological sample (*Mus musculus*) | macrophages | other | | defined as F4/80$^+$ or F4/80$^+$ CD11b$^+$ cells |
| biological sample (*Mus musculus*) | Treg cells | other | | defined as CD4$^+$ Foxp3$^+$ T cells |
| biological sample (*Mus musculus*) | Tconv cells | other | | defined as CD4$^+$ Foxp3$^-$ T cells |
| antibody | Ultra-LEAF Purified anti-mouse CD3ε (Armenian hamster monoclonal) | BioLegend | RRID:AB_11149115 | clone: 145–2C11 |
| antibody | Ultra-LEAF Purified anti-mouse CD28 (Syrian hamster monoclonal) | BioLegend | RRID:AB_11150408 | clone: 37.51 |
| antibody | anti-gp120 (mouse monoclonal) | Genentech | | control Ig in this paper PMID: 16530040 |
| antibody | anti-CRIg | Genentech | | clone: 14G8 (mouse monoclonal; PMID: 19017980); clone: 17C9 (rat monoclonal; PMID: 16530040) |

*Continued on next page*

*Continued*

| Reagent type (species) or resources | Designation | Source or reference | Identifiers | Additional information |
|---|---|---|---|---|
| antibody | anti-TGF-β 1, 2, 3 (mouse monoclonal) | R and D Systems | RRID:AB_357931 | clone: 1D11 |
| antibody | anti-CD16/CD32 (rat SD monoclonal) | BD Biosciences | RRID:AB_394656 | clone: 2.4G2 |
| antibody | anti-CRIg (mouse monoclonal) | this paper | | Biotinylated anti-CRIg (clone: 14G8) prepared by our laboratory |
| antibody | anti-CD45 (rat monoclonal) | BioLegend | RRID:AB_312981 | clone: 30-F11 |
| antibody | anti-CD45.1 (mouse monoclonal) | BioLegend | RRID:AB_893346 | clone: A20 |
| antibody | anti-CD45.2 (mouse monoclonal) | BioLegend | RRID:AB_389211 | clone: 104 |
| antibody | anti-TCRβ (Armenian hamster monoclonal) | BioLegend | RRID:AB_493344 | clone: H57-597 |
| antibody | anti-CD4 (rat monoclonal) | BioLegend | RRID:AB_312719; RRID:AB_312713; RRID:AB_312715 | clone: RM4-5 |
| antibody | anti-CD8α (rat monoclonal) | BioLegend | RRID:AB_312747; RRID:AB_312761 | clone: 53–6.7 |
| antibody | anti-Thy1.1 (mouse monoclonal) | BioLegend | RRID:AB_961437 | clone: OX-7 |
| antibody | anti-Thy1.2 (rat monoclonal) | BioLegend | RRID:AB_492888 | clone: 30-H12 |
| antibody | anti-Helios (Armenian hamster monoclonal) | BioLegend | RRID:AB_10660749 | clone: 22F6 |
| antibody | anti-ICOS (rat monoclonal) | ebioscience | RRID:AB_2573563 | clone: 7E.17G9 |
| antibody | anti-CD122 (rat monoclonal) | BioLegend | RRID:AB_313226 | clone: 5H4 |
| antibody | anti-CD25 (rat monoclonal) | BioLegend | RRID:AB_312857; RRID:AB_312865 | clone: PC61 |
| antibody | anti-CD69 (Armenian hamster monoclonal) | BioLegend | RRID:AB_2260065 | clone: H1.2F3 |
| antibody | anti-CDF4/80 (rat monoclonal) | BioLegend | RRID:AB_893481 | clone: BM8 |
| antibody | anti-CD11b (rat monoclonal) | BioLegend | RRID:AB_312791; RRID:AB_755986 | clone: M1/70 |
| antibody | anti-CD11c (Armenian hamster monoclonal) | BioLegend | RRID:AB_313777 | clone: N418 |
| antibody | anti-CD19 (rat monoclonal) | BioLegend | RRID:AB_313643 | clone: 6D5 |
| antibody | anti-NKp46 (rat monoclonal) | BioLegend | RRID:AB_2235755 | clone: 29A1.4 |
| antibody | anti-IL-17 (rat monoclonal) | BioLegend | RRID:AB_536018 | clone: TC11-18H10.1 |
| antibody | anti-IFN-γ (rat monoclonal) | BioLegend | RRID:AB_315402 | clone: XMG1.2 |

*Continued on next page*

*Continued*

| Reagent type (species) or resources | Designation | Source or reference | Identifiers | Additional information |
|---|---|---|---|---|
| antibody | anti-Foxp3 (rat monoclonal) | ebioscience | RRID:AB_1518812 | clone: FJK-16s |
| antibody | anti-phospho ZAP70/Syk$^{Tyr319/Tyr352}$ (mouse monoclonal) | ebioscience | RRID:AB_2572664 | clone: n3kobu5 |
| antibody | anti-phospho ERK1/2$^{Thr202/Tyr204}$ (mouse monoclonal) | BioLegend | RRID:AB_2629710 | clone: 6B8B69 |
| antibody | anti-phospho AKT1$^{Ser473}$ (mouse monoclonal) | ebioscience | RRID:AB_2573309 | clone: SDRNR |
| antibody | anti-phospho S6$^{Ser235, Ser236}$ (mouse monoclonal) | ebioscience | RRID:AB_2572666 | clone: cupk43k |
| antibody | anti-phospho STAT5 (mouse monoclonal) | BD Biosciences | RRID:AB_10894188 | clone: 47/Stat5(pY694) |
| antibody | anti-phospho Smad2 (pS465/pS467)/ Smad3 (pS423/pS425) (mouse monoclonal) | BD Biosciences | RRID:AB_2716578 | clone: O72-670 |
| antibody | PerCP/Cy5.5-streptavidin | BioLegend | RRID:AB_2716577 | |
| recombinant DNA reagent | pCR 2.1-TOPO (vector) | Invitrogen | CAT#: K204040 | |
| sequence-based reagent | Foxp3 Intron1 | Eton Bioscience | | Forward: ATTTGAATTGGATATGGTTTGT; Reverse: AACCTTAAACCCCTCTAACATC |
| sequence-based reagent | Foxp3 TSDR | Eton Bioscience | | Forward: GTTTGTGTTTTTGAGATTTTAAAAT; Reverse: AACCAACTTCCTACACTATCTATTA |
| sequence-based reagent | Rara | Eton Bioscience | | Forward: CCAGTCAGTGGTTACAGCACA; Reverse: TAGTGGTAGCCGGATGATTTG |
| sequence-based reagent | Rarb | Eton Bioscience | | Forward: ACATGATCTACACTTGCCATCG; Reverse: TGAAGGCTCCTTCTTTTTCTTG |
| sequence-based reagent | Rarg | Eton Bioscience | | Forward: CATTTGAGATGCTGAGCCCTA; Reverse: GCTTATAGACCCGAGGAGGTG |
| sequence-based reagent | Oligo(dT)$_{12-18}$ Primer | Invitrogen | CAT#: 18418012 | |
| peptide, recombinant protein | CRIg-Ig | Genentech | | PMID: 16530040 |
| peptide, recombinant protein | Biotinylated CRIg-Ig | this paper | | prepared by our laboratory |
| peptide, recombinant protein | control Ig | Genentech | | PMID: 16530040 |
| peptide, recombinant protein | Biotinylated control Ig | this paper | | prepared by our laboratory |
| peptide, recombinant protein | CTLA-4 Ig | R and D Systems | CAT#: 434-CT-200/CF | |
| peptide, recombinant protein | PD-1 Ig | R and D Systems | CAT#: 1021-PD-100 | |
| peptide, recombinant protein | VISTA Ig | R and D Systems | CAT#: 7005-B7-050 | |

*Continued on next page*

*Continued*

| Reagent type (species) or resources | Designation | Source or reference | Identifiers | Additional information |
|---|---|---|---|---|
| peptide, recombinant protein | CD226 Ig | R and D Systems | CAT#: 4436-DN-050 | |
| peptide, recombinant protein | TIGIT Ig | R and D Systems | CAT#: 7267-TG-050 | |
| peptide, recombinant protein | BDC2.5 mimotope | AnaSpec | CAT#: AS-63774 | Sequence: RTRPLWVRME |
| peptide, recombinant protein | Recombinant Murine IL-2 | PeproTech | CAT#: 212–12 | |
| peptide, recombinant protein | Recombinant Human TGF-β1 | PeproTech | CAT#: 100–21 | |
| commercial assay or kit | ACK lysing buffer | Lonza | CAT#: 10-548E | |
| commercial assay or kit | Anti-PE microBeads | Miltenyi Biotec | CAT#: 130-048-801 | |
| commercial assay or kit | LIVE/DEAD fixable dead cell stain kits | Invitrogen | CAT#: L34972; CAT#: L34966 | |
| commercial assay or kit | Foxp3/Transcription factor staining buffer set | ebioscience | CAT#: 00-5523-00 | |
| commercial assay or kit | Phosflow Lyse/Fix buffer | BD Biosciences | CAT#: 558049 | |
| commercial assay or kit | Phosflow Perm buffer III | BD Biosciences | CAT#: 558050 | |
| commercial assay or kit | Percoll | GE Healthcare Life Science | CAT#: 17-0891-01 | |
| commercial assay or kit | NucleoSpin Tissue XS | Macherey-Nagel | CAT#: 740901.50 | |
| commercial assay or kit | EZ DNA Methylaiton Kit | Zymo Research | CAT#: D5001 | |
| commercial assay or kit | HotStarTaq DNA Polymerase | QIAGEN | CAT#: 203203 | |
| commercial assay or kit | TOPO TA Cloning Kit | Invitrogen | CAT#: K204040 | |
| commercial assay or kit | TRIzol Reagent | Invitrogen | CAT#: 15596026 | |
| commercial assay or kit | SuperScript III Reverse Transcriptase | Invitrogen | CAT#: 18080044 | |
| commercial assay or kit | SYBR Green PCR Master Mix | Applied Biosystems | CAT#: 4309155 | |
| chemical compound, drug | Collagenase P | Roche | CAT#: 11249002001 | |
| chemical compound, drug | Collagenase D | Roche | CAT#: 11088882001 | |
| chemical compound, drug | DNase I | Sigma-Aldrich | CAT#: DN25-1G | |
| chemical compound, drug | ATRA | Sigma-Aldrich | CAT#: R2625-50MG | |
| chemical compound, drug | BMS 493 | Sigma-Aldrich | CAT#: B6688-5MG | |

*Continued*

| Reagent type (species) or resources | Designation | Source or reference | Identifiers | Additional information |
|---|---|---|---|---|
| chemical compound, drug | Vancomycin | Acros Organics | CAT#: 296990010 | |
| chemical compound, drug | Metronidazole | Acros Organics | CAT#: 210340050 | |
| chemical compound, drug | Neomycin | Fisher Scientific | CAT#: BP266925 | |
| chemical compound, drug | Ampicillin | Sigma-Aldrich | CAT#: A0166-25G | |
| chemical compound, drug | PMA | Sigma-Aldrich | CAT#: P1585-1MG | |
| chemical compound, drug | Ionomycin | Sigma-Aldrich | CAT#: I0634-1MG | |
| chemical compound, drug | Brefeldin A solution | BioLegend | CAT#: 420601 | |
| chemical compound, drug | Fisher Healthcare Tissue-Plus O.C.T Compound | Fisher Scientific | CAT#: 23-730-571 | |
| chemical compound, drug | Avidin, HRP conjugate | Invitrogen | CAT#: 434423 | |
| chemical compound, drug | 1-Step Ultra TMB-ELISA Substrate Solution | Thermo Scientific | CAT#: 34028 | |
| chemical compound, drug | Stop Solution for TMB Substrates | Thermo Scientific | CAT#: N600 | |
| chemical compound, drug | DAPI | Invitrogen | CAT#: D1306 | |
| software, algorithm | FlowJo | FlowJo, LLC | RRID:SCR_008520 | |
| software, algorithm | ImageJ | NIH | RRID:SCR_003070 | |
| software, algorithm | BISMA software | other | RRID:SCR_000688 | public website, BDPC DNA methylation analysis platform |

## Mice

B6 and NOD/Foxp3$^{GFP}$ mice (*Haribhai et al., 2007*) were purchased from Jackson laboratory. NOD and NOD/BDC2.5/Thy1.1 mouse lines were obtained from Mathis-Benoist laboratory (*Hill et al., 2007*). NOD/BDC2.5/Thy1.1 mice were further crossed to NOD/Foxp3$^{GFP}$ to generate NOD/BDC2.5/Foxp3$^{GFP}$/Thy1.1$^+$ and NOD/BDC2.5/Foxp3$^{GFP}$ Thy1.1$^+$ Thy1.2$^+$ mice. B6/CRIg KO mice were provided by Genentech (*Helmy et al., 2006*). NOD/CRIg KO mice were generated by crossing B6/CRIg KO mice onto a NOD background for more than 10 generations. Speed-congenics was used to facilitate the generation of NOD/CRIg KO mice. All mice were housed under specific pathogen free (SPF) conditions in our animal facility at University of California, San Diego, in accordance with the ethical guidelines of the Institutional Animal Care and Use Committee.

## Cell preparations

The pancreas was minced and digested in 10 ml phenol red-free Dulbecco's modified Eagle's medium (DMEM) containing collagenase P (0.5 mg/ml) (Roche, Indianapolis, IN) at 37°C on a 200 rpm shaker for 20 min. After digestion, single cells were filtered by passing through a 70 um cell strainer and suspended in FACS staining buffer. Colon lamina propria (cLP) cells were prepared as previously described. Briefly, the intestine was cut longitudinally and sliced into 0.2–0.5 cm pieces, followed by incubating in 10 ml Ca2$^+$/Mg2$^+$ free HBSS containing 10% FBS, 15 mM HEPES, and 5 mM EDTA at 37°C on a 200 rpm shaker for 20 min. The intestine fragments were further digested in 10 ml RPMI 1640 containing 5% FBS, Collagenase D (0.5 mg/ml) (Roche) and DNase I (0.05 mg/ml)

(Sigma-Aldrich, Saint Louis, MO) at 37°C on a 200 rpm shaker for 45 min. Samples were collected by passing through a 70 um cell strainer. cLP cells were then purified on a 44/67% Percoll (GE Health-care Life Science, Chicago, IL) gradient by centrifugation (800 x g, 20 min at room temperature). Single-cell suspensions of lymphoid organs were prepared by mechanic disruption. Red blood cells were lysated with Ammonium-Chloride-Potassium (ACK) buffer (Lonza, Walkersville, MD). Peritoneal cells were collected by flushing 1 x PBS into the peritoneal cavity.

### In vitro T cell activation

For T cell activation assays, 96-well tissue-culture plates were pre-coated with anti-CD3 (2.5 ug/ml) and CRIg-Ig (5 ug/ml), or control Ig in 1 x PBS at 4°C overnight. The plates were washed with 1 x PBS three times. Single cell suspensions from spleen, peripheral lymph node were prepared. $CD4^+$ $CD25^-$ cells were MACS (Miltenyi Biotec, Auburn, CA) enriched and labeled with CFSE or CTV. In some experiments, $CD4^+$ $Foxp3(GFP)^-$ Tconv cells were FACS sorted. Isolated Tconv cells ($1 \times 10^5$/well) were cultured in RPMI1640 supplemented with 10% FBS, L-glutamine (2 mM), penicillin (100 U/ml), streptomycin (100 ug/ml), and non-essential amino acids (100 Um) and β-mercaptoethanol (50 uM), in the presence of soluble anti-CD28 (2.5 ug/ml). CRIg-Ig fusion proteins were pre-coated or added as soluble form as indicated. In some experiments, anti-CRIg mAb (clone 17C9, or 14G8) was added to the culture either in a soluble form, or pre-coated as indicated. To test the effect of cross-linking of CRIg by anti-CRIg mAb, plates were pre-coated with different concentrations of anti-CRIg mAb (clone 17C9), or isotype control mAb, and anti-CD3. Soluble anti-CD28 and soluble CRIg-Ig (5 ug/ml) were added to the cultured cells.

### In vitro iTreg differentiation and restimulation

96-well tissue-culture plates were pre-coated with either CRIg-Ig (2.5 ug/ml), or control Ig. $CD4^+$ $Foxp3^-$ Tconv cells were sorted from NOD/$Foxp3^{GFP}$ reporter mice, labeled with CFSE or CTV and cultured ($1 \times 10^5$) in the presence of plate-bound anti-CD3 (2.5 ug/ml), soluble anti-CD28 (2.5 ug/ml), murine IL-2 (100 U/ml) and TGF-β (1 ng/ml or as indicated) for 3 days. In some experiments, an anti-TGF-β neutralizing antibody (clone 1D11) was added. In antigen-stimulated Treg differentiation assays, the plates were pre-coated with either CRIg-Ig (10 ug/ml), or control Ig. Total splenocytes ($2 \times 10^5$) from NOD/BDC2.5/Thy1.1 mice were labeled with CTV and cultured with mimotope (100 ng/ml) (AnaSpec, Fremont, CA) for 3 days. Cells were then analyzed for the expression of GFP or intra-cellular staining for Foxp3.

For the culture of restimulated iTreg cells, iTreg cells were generated by TGF-β, without CRIg-Ig. After 24 hr, GFP $(Foxp3)^+$ cells were sorted and restimulated under the condition of plate-bound anti-CD3 (2.5 ug/ml), soluble anti-CD28 (2.5 ug/ml), IL-2 (100 U/ml), anti-TGF-β neutralizing antibody (clone 1D11), with plate-bound CRIg-Ig (5 ug/ml) or control Ig, respectively. After 3 days, cultured cells were collected and analyzed for the expression of GFP or intracellular staining for Foxp3. In some experiments, purified Tconv cells were activated for 24 hr, washed and transferred to a new plate with coated anti-CD3 (2.5 ug/ml), soluble anti-CD28 (2.5 ug/ml), IL-2 (100 U/ml) and TGF-β (1 ng/ml) and cultured for 3 days.

For in vitro Treg suppression assay, iTreg cells were generated as above with CRIg-Ig, or Control-Ig, respectively. After 3 days, GFP $(Foxp3)^+$ cells were sorted and co-cultured with MACS enriched CTV labeled $CD4^+CD25^-$ cells at the ratio of 1:4, 1:2 and 1:1, respectively, in the presence of plate-bound anti-CD3, soluble anti-CD28 for 3 days.

### In vitro macrophage culture

$F4/80^+$ macrophages from peritoneal cavity were sorted as $CRIg^+$ or $CRIg^-$ subsets and cultured in complete DMEM in the presence of RA (100 nM) (Sigma-Aldrich) or control DMSO. After 3 days of culture, adherent macrophages were harvested by adding Trypsin and EDTA (5 mM) and analyzed for CRIg expression by FACS.

### Adoptive cell transfer and in vivo treatments

For in vivo CRIg-Ig treatment, 10-week-old pre-diabetic NOD mice were *i.p.* injected with CRIg-Ig (3.5 mg/kg) mixed with anti-CRIg mAb (17C9) (7 mg/kg) or control Ig (3.5 mg/kg), twice a week for 2 weeks. Pancreas and other lymphoid organs were analyzed after the treatment. For in vivo pTreg

induction assays, CD4$^+$ Foxp3(GFP)$^-$ Tconv cells were sorted from NOD/BDC2.5/Foxp3$^{GFP}$/Thy1.1$^+$ mice and *i.p.* transferred to 4-week-old NOD or NOD/CRIg KO mice. In some experiments, NOD mice received adoptively transferred Tconv cells were injected with CRIg-Ig, or control Ig every other day for 2 weeks. After 2 weeks, transferred cells collected from the pancreas and lymphoid organs were analyzed. For in vivo iTreg stability assays, CD4$^+$ Foxp3(GFP)$^-$ Tconv cells were sorted from NOD/Foxp3$^{GFP}$/Thy1.1$^+$ mice, or NOD/Foxp3$^{GFP}$/Thy1.1$^+$ Thy1.2$^+$ mice, and differentiated into iTreg cells under conditions with plate-bound CRIg-Ig and control Ig respectively. iTreg cells from these two conditions were sorted and mixed (2 × 10$^5$ each) at 1:1 ratio before transferring into 10–14 days old NOD mice (Thy1.2$^+$). One week later, transferred cells from spleen and pancreatic islets were isolated and analyzed by flow cytometry. For antibiotics treatment, 8-week-old B6, or 10-week-old NOD mice were given a combination of four antibiotics (vancomycin, metronidazole, neomycin and ampicillin, VMNA) (*Sefik et al., 2015*) in drinking water supplement with sugar. Peritoneal cavity, pancreatic and colonic macrophages were analyzed one week later. For in vivo treatment of BMS 493 (Sigma-Aldrich), 7 weeks old B6 mice were *i.p.* injected with BMS 493 (100 nM) daily for 3 days. In some experiments, peritoneal cavity macrophages (3 × 10$^4$) were sorted from B6/CD45.1 mice and *i.p.* transferred into B6/CD45.2 mice, followed by BMS 493 treatment. Peritoneal and pancreatic macrophages were analyzed after the last injection.

## Abs and flow cytometry

All stainings began with an incubation with a mAb for CD16/32 (2.4G2 BD Biosciences). mAbs to indicated molecules used in this study were: Biotin-CRIg were prepared by our laboratory. CD45 (30-F11), CD45.1 (A20), CD45.2 (104), TCRβ (H57-597), CD4 (RM4-5), CD8α (53–6.7), Thy1.1 (OX-7), Thy1.2 (30-H12), Helios (22F6), CD122 (5H4), CD25 (PC61), CD69 (H1.2F3), F4/80 (BM8), CD11b (M1/70), CD11c (N418), CD19 (6D5), NKp46 (29A1.4), IL-17 (TC11-18H10.1), and IFN-γ (XMG1.2) from BioLegend (San Diego, CA); ICOS (7E.17G9) and Foxp3 (FJK-16s) from eBioscience (San Diego, CA) . PerCP/Cy5.5-streptavidin was from BioLegend. Intracellular staining of Foxp3 and Helios was performed using Foxp3/transcription factor staining buffer set (eBioscience) according to the manufacturer's instructions. For intracellular IFN-γ and IL-17 detection, unfractionated cells were cultured in complete medium in the presence of phorbol myristate acetate (PMA), and Ionomycin (Signa-Aldrich) and Brefeldin A (BioLegend) at 37°C for 4 hr. Samples were acquired with a BD LSRFortessa (BD, San Jose, CA) and analyzed with FlowJo software (FlowJo, LLC, Ashland, OR).

## Binding assays

CD4$^+$ CD25$^-$ Tconv cells from 10-week-old NOD mice, or CD4$^+$ GFP(Foxp3)$^+$ Treg cells from 10-week-old NOD/Foxp3$^{GFP}$ mice, were cultured under the in vitro T cell activation condition. At indicated time-points, different concentrations of biotinylated CRIg-Ig or control Ig were incubated with cultured T cells at 37°C for 1 hr. Cells were then harvested and stained with Fixable Viability Dyes (Invitrogen, Carlsbad, CA) before fixed with paraformaldehyde (1.6%). Streptavidin-conjugated secondary Abs were added to measure the binding intensities of biotinylated CRIg-Ig or control Ig.

## Enzyme-linked immunosorbent assay

Flat bottom Costar 96-well plate were coated with 100 ul rat anti-mouse CRIg (clone 19E4, 1 ug/ml) in 0.05 M Na$_2$CO$_3$ pH 9.6 at 4°C overnight. Coated Plates were washed with 1 x PBS and blocked with blocking buffer which contains 50 mM Tris, 0.14 M NaCl, and 1% BSA at 37°C for 2 hr. Mice sera were diluted (1:20) and 100 ul of sample was added to each well and incubated at 37°C for 2 hr. After incubation, plates were washed with 1x PBS containing 50 mM Tris, 0.14 M NaCl, 0.05% Tween 20, and incubated with 100 ul 1 ug/ml biotinylated anti-CRIg mAb (14G8) in blocking buffer at 37°C for 1 hr. After washing, plates were then incubated with diluted HRP-Avidin (1: 20,000) (Invitrogen) at 37°C for 1 hr. Plates were washed and incubated with TMB substrate at 37°C for 30 min. Stop solution was added before plates were read on microplate reader at an optical density (O. D.) of 450 nm (reference 570 nm).

## Phosphorylation assays

CD4$^+$ CD25$^-$ cells were cultured under the in vitro T cell activation conditions. At indicated time pointes, the cells were fixed with paraformaldehyde (1.6%) for 10 min at 37°C. Fixed cells were

permeabilized with ice-cold methanol for 30 min, and stained with the phospho Abs for ZAP70[Tyr319/Tyr352] (clone n3kobu5; eBioscience), ERK1/2[Thr202/Tyr204] (clone 6B8B69; BioLegend), AKT1[Ser473] (clone SDRNR; eBioscience) and S6[Ser235/236](clone cupk43k; eBioscience) for 1 hr at room temperature. In Treg phosphorylation assays, control or CRIg iTreg cells were prepared as above described. After 18 hr, iTreg cells were fixed, permeabilized, and stained with phospho Abs for Akt[Ser473], S6[Ser235/236]. For the analysis of phospho-STAT5, iTreg cells first rested in complete RPMI1640 medium for 30 min and then incubated with murine IL-2 (100 U/ml) for 15 min at 37°C. After that, the cells were fixed and permeabilized as described above and subsequently stained with anti-phospho-STAT5 (clone 47/Stat5(pY694); BD Biosciences) for 1 hr at room temperature. To detect the phosphor Smad2/3, total splenocytes from NOD mice were serum-starved overnight. CD4$^+$ CD25$^-$ cells were then sorted and cultured in medium with 0.2% of fetal bovine serum under iTreg differentiation condition. At indicated time-points, cultured T cells were fixed in Phosflow Lyse/Fix Buffer (BD Biosciences) for 10 min at 37°C and permeabilized in Phosflow Perm Buffer III (BD Biosciences) for 30 min on ice. Cells were then stained with mouse anti-Smad2 (pS465/pS467)/Smad3 (pS423/pS425) (O72/670; BD Biosciences) mAb for 1 hr at room temperature.

## Immunostaining and histology

Tissue samples were cut and snap frozen in optimum cutting temperature (O.C.T., Fisher Scientific, Houston, TX). 6 um cryo-sections of tissue sections were cut and fixed with pre-cold acetone for 20 min. Immunostaining was performed as previously described (*Fu et al., 2014*). Before adding primary mAbs, sections were blocked with 5% normal donkey serum (Jackson ImmunoResearch, West Grove, PA). The following mAbs were used in different combinations as indicated in the figure legends: anti-CRIg mAb (a gift from Genentech, South San Francisco, CA); anti-F4/80 (BM8, BioLegend); anti-CD4 (RM4-5, BioLegend). Nuclei were stained with DAPI (4',6-Diamidino-2–28 phenylindole dihydrochloride) (Invitrogen). Images were acquired on an AxioImager microscope (Zeiss, Thornwood, NY), and were processed with ImageJ (NIH). For histology assays, the pancreases were removed and fixed with 10% formalin solution. Fixed tissue blocks were paraffin-embedded, sectioned, and stained with haematoxylin and eosin.

## Bisulfite sequencing

Genomic DNA was extracted by using NucleoSpin Tissue XS (Macherey-Nagel, Bethlehem, PA) and bisulfite converted by using the EZ DNA Methylation Kit (Zymo Research, Irvine, CA). Foxp3 CNS2 region was amplified by PCR containing 5 ng of bisulfite-converted genomic DNA, HotStar Taq PCR buffer (QIAGEN, Valencia, CA), 0.5 U HotStar Taq DNA polymerase, 2.5 mM MgCl$_2$ and 0.38 μM each of forward and reverse primers (*Ohkura et al., 2012*; *Yang et al., 2016*) in a final volume of 25 μl (95°C for 10 min; 40 cycles: 95°C for 30 s, 58°C for 1 min, 72°C for 1 min; 72°C for 5 min). The PCR product was analyzed by gel electrophoresis and TA cloned using TOPO TA Cloning Kit (Invitrogen) and One Shot TOP10 Chemically Competent E. coli. Colonies were picked and submitted for direct colony sequencing. Sequencing results were analyzed on Bisulfite Sequencing DNA Methylation Analysis website (*Rohde et al., 2010*).

## Real time RT-PCR

CRIg$^+$ and CRIg$^-$ macrophages from the peritoneal cavity of 7-week-old B6 mice were FACS sorted. Total RNA was extracted using Trizol and reverse-transcribed using oligo(dT) and SuperScript III (Invitrogen). PCR was performed using Power SYBR Green PCR Master Mix (Applied Biosystems, Foster City, CA), using CFX96 Touch Real-Time PCR Detection System (Bio-Rad, Hercules, CA) for PCR and signal detection. The primers were: Rara forward: ccagtcagtggttacagcaca. Rara reverse: tagtggtagccggatgatttg. Rarb forward: acatgatctacacttgccatcg. Rarb reverse: tgaaggctccttctttttcttg. Rarg forward: catttgagatgctgagcccta. Rarg reverse: gcttatagacccgaggaggtg.

## Statistics

Student's t-test was used for statistical analyses. *P* values of 0.05 or less were considered statistically significant.

## Acknowledgements

We thank Dr. M. van Lookeren Campagne (Genentech) for insightful discussions and for kindly providing us with CRIg KO mice, monoclonal antibodies to CRIg and the CRIg-Ig fusion proteins. We thank Dr. Li-Fan Lu (UCSD) for providing us with cells from B6/CD45.1 mice. The human pancreatic frozen sections were kindly provided by Dr. Jerrold Olefsky (UCSD). We also thank J Olvera and C Fine for help with cell sorting. This work was supported by the AAI Career in Immunology Fellowship (to XY), NIDDK P30 DK063491, UCSD CTRI UL1 TR000100 and JDRF 2-SRA-2016–306 s-B (to WF).

## Additional information

### Funding

| Funder | Grant reference number | Author |
| --- | --- | --- |
| American Association of Immunologists | | Xiaomei Yuan |
| JDRF | 2-SRA-2016-306-S-B | Wenxian Fu |
| National Institute of Diabetes and Digestive and Kidney Diseases | P30 DK063491 | Wenxian Fu |
| University of California, San Diego | UL1 TR000100 | Wenxian Fu |

The funders had no role in study design, data collection and interpretation, or the decision to submit the work for publication.

### Author contributions

Xiaomei Yuan, Conceptualization, Data curation, Formal analysis, Validation, Investigation, Methodology, Writing—original draft, Writing—review and editing; Bi-Huei Yang, Yi Dong, Investigation, Methodology; Asami Yamamura, Investigation; Wenxian Fu, Conceptualization, Resources, Data curation, Formal analysis, Supervision, Funding acquisition, Validation, Investigation, Visualization, Methodology, Writing—original draft, Project administration, Writing—review and editing

### Author ORCIDs

Wenxian Fu http://orcid.org/0000-0003-0684-8929

### Ethics

Animal experimentation: All mice were housed under specific pathogen free (SPF) conditions in our animal facility at University of California, San Diego, in accordance with the ethical guidelines of the Institutional Animal Care and Use Committee (#S13253).

### Decision letter and Author response

Decision letter https://doi.org/10.7554/eLife.29540.027
Author response https://doi.org/10.7554/eLife.29540.028

## Additional files

### Supplementary files

• Transparent reporting form
DOI: https://doi.org/10.7554/eLife.29540.024

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
