## [Decision Letter]

Thank you for submitting your article "CRIg, a tissue-resident macrophage specific coinhibitory molecule, promotes immunological tolerance via a dual role in effector and regulatory T cells" for consideration by *eLife*. Your article has been reviewed by three peer reviewers, one of whom is a member of our Board of Reviewing Editors, and the evaluation has been overseen by Tadatsugu Taniguchi as the Senior Editor. The reviewers have opted to remain anonymous.

The reviewers have discussed the reviews with one another and the Reviewing Editor has drafted this decision to help you prepare a revised submission.

Summary:

The manuscript by Yuan, et al. describes a role of the complement receptor of immunoglobulin family (CRIg), which is exclusively expressed by tissue macrophages, in the induction of peripheral Foxp3-expressing Treg cells. The authors demonstrated that CRIg directly acted on an unknown ligand/receptor of activated T cells, promoting and stabilizing the differentiation of Treg cells via regulating IL-2 signaling pathways. Micro-environmental cues in the gut, including microbiota and metabolites (retinoic acid), were shown to control the expression of CRIg on tissue macrophages, leading to the induction of Treg-mediated immunological tolerance.

Essential revisions:

1) It has been reported that CRIg plays a key role in tolerance induction. One major advance claimed by the paper is the promotion and stabilization of Foxp3^+^ peripheral Tregs by CRIg. However, the authors mainly used a CRIg-Ig fusion protein in their experiments to induce Treg cells and tolerance, which may be artificial. They should purify CRIg^+^ macrophages and coculture them with activated Th cells to test whether the macrophages with physiological levels of CRIg can specifically induce Foxp3^+^ Tregs.

2) It was shown that CRIg-Ig promoted iTreg differentiation in vitro, but the authors should address whether CRIg^+^ macrophages could directly induce pTregs in in vivo settings. This will strengthen their claims. The frequency of Tregs in the pancreas was reduced in CRIg^-/-^ NOD as shown in Figure 1. However, this experimental system does not exclude a possibility of indirect contribution of CRIg to the expansion of pTregs.

In addition, regarding Figure 1, how did the authors investigate the percentages of CRIg^+^ macrophages and each T cell population in pancreas? Additionally, the authors seemed to isolate cells from pancreatic islets selectively, but how did they do this? There seems to be no explanation for these procedures in the manuscript.

3) Relating to the above point, where do pancreatic CRIg^+^ macrophages interact with activated Tconv cells and convert them into pTregs? Is there co-localization of CRIg^+^ macrophages and converted Foxp3^+^ Tregs in the pancreas?

4) In Figure 2, CRIg-Ig binding to activated Tconv is shown, but staining intensity is marginal and not convincing. More kinetics and control cells should be taken. In addition, expression levels of CRIg on thymic, peripheral, and in vitro induced Tregs need to be examined.

5) With CRIg^-/-^ BL6 mice, are the findings in NOD mice also the case in BL6 mice? Since macrophages from various tissues express CRIg, the authors should examine whether CRIg^-/-^ mice have reduced frequencies of Tregs in mucosal tissues such as the gut, peritoneum, skin, and lung, where CRIg^+^ tissue macrophages are present.

6) The rationale for the CRIg/C3 blockade experiments in Figure 2 supplements. is unclear. A combination of CRIg-Ig and anti-CRIg completely blocked the proliferation of T conv cells. Although the authors assume that anti-CRIg can block the binding of CRIg to C3 components, this unexplained strong effect of anti-CRIg on T cell suppression is a bit odd. Are active C3 components present in this culture medium? Why does anti-CRIg induce a sort of super agonistic effect of CRIg? On the other hand, this complete T cell proliferation arrest (Figure 2—figure supplement 1) with CRIg and anti-CRIg contradicts with other results (Figure 6—figure supplement 1) in which T cells robustly proliferate in the presence of CRIg and anti-CRIg.

7) In Figure 3, the differentiation of pTregs from Tconv was impaired in the pancreas, but not in LNs of CRIg^-/-^ NOD mice which had received adoptive transfer of BDC2.5 Tconv. This is a bit odd. It is unlikely that there is enough cellular niche to allow Tconv to get activated, expand, and differentiate into pTreg in T cell-sufficient NOD mice. They should show more detailed FACS profiles, for example, the proportion of Thy1.1 (transferred) vs Thy1.2 (resident) in Foxp3^+^ Tregs in the pancreas and LNs.

Also, CRIg-Ig treatment enhanced pTreg generation in pancreas but not in panLNs and ILNs in Figure 3. The authors should discuss the reason for this result.

8) Is the target molecule of CRIg on effector T cells and regulatory T cells same or different? The authors should discuss how CRIg exhibit opposite effects on the two different T cell populations.

9) It is hard to understand why the authors focused on environmental factors that modulate CRIg expression on TRMs in the tissues other than pancreas. Is there any relationship between CRIg levels in pancreatic TRMs and bacterial colonization or RA? How about the expression of CRIg in pancreatic TRMs after birth in wild-type and NOD mice? It is more valuable for this study to find the environmental cues that would regulate CRIg levels in pancreatic TRMs, if possible, especially during the development of type 1 diabetes.

10) Does CRIg bind to Treg and synergize with TGFb? Binding of CRIg to Tregs should be added (as in Figure 2 for activated T cells) along with key controls.

11) Treg stability depends on the methylation status of FoxP3 CNS2 region. The authors have shown that CRIg does not affect the methylation status of CNS2 site of Foxp3 yet FoxP3 expression in adoptively transferred iTregs generated with CRIg-Ig is remarkably stable. The inference from this observation is that Tregs do not need continuous cell:cell contact with CRIg-expressing macrophages to maintain FoxP3 expression and function. The authors should discuss this point.

12) The experiments in Figure 6 and Figure 7 are interesting and compelling. However they do not connect well with the rest of the paper. These features should have been examined in pancreatic macrophages to make Figure 6 and Figure 7 cohesive with Figure 1-5.

---

## [Author Response]

Essential revisions:1) It has been reported that CRIg plays a key role in tolerance induction. One major advance claimed by the paper is the promotion and stabilization of Foxp3^+^ peripheral Tregs by CRIg. However, the authors mainly used a CRIg-Ig fusion protein in their experiments to induce Treg cells and tolerance, which may be artificial. They should purify CRIg^+^macrophages and coculture them with activated Th cells to test whether the macrophages with physiological levels of CRIg can specifically induce Foxp3^+^ Tregs.

We thank the reviewer for the suggestion. We have performed the suggested experiments and included the data as Figure 3. Our results showed that coculturing with CRIg^+^ macrophages enhanced the generation of iTreg cells in vitro, compared with CRIg^-^ macrophages.

However, we also wanted to emphasize that in this study, we focus on understanding the role of CRIg molecule. As CRIg^+^ macrophages may have other functions that are independent of CRIg.

2) It was shown that CRIg-Ig promoted iTreg differentiation in vitro, but the authors should address whether CRIg^+^macrophages could directly induce pTregs in in vivo settings. This will strengthen their claims. The frequency of Tregs in the pancreas was reduced in CRIg^-/-^NOD as shown in Figure 1. However, this experimental system does not exclude a possibility of indirect contribution of CRIg to the expansion of pTregs.

We understand that the reviewer wanted to see whether there was a direct effect of CRIg^+^ macrophages in Treg induction. We could isolate and transfer CRIg^+^ macrophages into congenically disparate recipient mice. However, we do not think this assay could address whether these transferred CRIg^+^ macrophages expand pTreg cells directly or by influencing other cells or pathways. Another caveat is that it is challenging to assess tissue engraftment of transferred macrophages, as these cells do not express CCR2 and other key chemokine receptors for tissue migration. We think that an in vitro coculture could define the direct effect of these TRMs in iTreg induction. We have performed this experiment, as it also answered the question raised in comment #1. We have included these data as Figure 3.

We, however, propose that genetic ablation of CRIg would provide critical clues of the role of CRIg in affecting Treg (especially pTreg) abundancy, and our results supported this notion. To further strengthen this claim, we now have specifically analyzed the proportion of Helios^-^ Tregs (largely pTregs) in various tissue sites (pancreas, colon and lung) of NOD/CRIg^-/-^and wildtype littermate controls. CRIg^+^ macrophages are present in pancreas and colon, but not in the lung (Figure 1—figure supplement 1). Our data showed that Helios^-^ pTregs in pancreas and colon, but not lung, were preferentially reduced in CRIg null mutant mice, correlated with the tissue-distribution profile of CRIg^+^ macrophages. These data further supported the role of CRIg in affecting the abundancy of pTreg. We have now included these pTreg results between CRIg^-/-^and wildtype mice as Figure 3, and added the analyses of CRIg^+^ cells in the lung in Figure 1—figure supplement 1.

In addition, regarding Figure 1, how did the authors investigate the percentages of CRIg^+^macrophages and each T cell population in pancreas? Additionally, the authors seemed to isolate cells from pancreatic islets selectively, but how did they do this? There seems to be no explanation for these procedures in the manuscript.

We apologize for any ambiguity in the descriptions.

For CRIg^+^ macrophages, the numbers showed in Figure 1 (right panel) were counted from immunostaining of pancreatic sections of 10-wks old NOD mice. We examined each individual islet under microscope and counted the total number of CRIg^+^ TRMs per each islet. For this measurement, we did not isolate these cells from pancreatic islets. We have now updated the descriptions to improve the clarity in the Materials and methods, figure legends and text. We here wanted to explain a bit more about why we chose such a method. Because during the development of insulitis in both mice (e.g., NOD strain) and humans, not all islets are synchronously infiltrated (there are intact islets even in long-standing diabetic patients) (Bluestone, Herold and Eisenbarth, 2010), the reasons for this heterogeneous penetrance remain poorly understood. Due to the unique peri-islet distribution of CRIg^+^ TRMs and the function of CRIg, our studies led to a model whereby CRIg^+^ macrophages form a protective barrier for each individual islet. This provided a plausible explanation to the long-standing question of why some islets are infiltrated while others are not. The dearth of CRIg^+^ TRMs at a barrier of the islet may exacerbate the infiltration.

For T cells, the “% Treg in CD4” in panel C was calculated by gating on Foxp3 fractions within CD4^+^ populations. It also reflected the ratio between Treg and Foxp3^-^ CD4 T cells. For the panel of “Ratio Treg/CD8”, it meant within the pancreas of each individual mouse (depicted by each dot), the number of Treg cells divided by the number of CD8^+^ T cells. Using this analysis, it clearly highlighted the loss of balance between Treg cells and other T cells that are largely pathogenic. We have now updated the descriptions in the Materials and methods, figure legends and text for clarity.

3) Relating to the above point, where do pancreatic CRIg^+^macrophages interact with activated Tconv cells and convert them into pTregs? Is there co-localization of CRIg^+^macrophages and converted Foxp3^+^ Tregs in the pancreas?

We thank the reviewers for raising this question. We performed a co-staining of CRIg and CD4 of pancreatic frozen sections from a 10-wk old NOD mice (having ongoing insulitis) to reveal whether there was cell-cell contact between CRIg^+^ TRMs and CD4 T cells. Indeed, we found cell aggregates containing both CRIg^+^ TRMs and CD4^+^ T cells, conforming the close interactions among them. We have now included the results as Figure 1—figure supplement 3.

Again, these data supported our model whereby CRIg^+^ macrophages form a protective barrier for each individual islet. Notably, while we were revising this manuscript, a paper published by Mohan et al. (Mohan JF, PNAS 2017. 114: E7776) suggested a similar role of peri-islet macrophages. We quote their statements here and note that one graph from their study depicts the localizations of macrophages and T cells in pancreas of NOD mice; “Suppressive roles of MFs have also been reported (Fu et al., 2012), which may be accounted for by their position as gate-keepers to islet entry in the initial phases, or by the capsule that MFs appear to form around the remaining β-cell mass in cases of established insulitis”.

4) In Figure 2, CRIg-Ig binding to activated Tconv is shown, but staining intensity is marginal and not convincing. More kinetics and control cells should be taken. In addition, expression levels of CRIg on thymic, peripheral, and in vitro induced Tregs need to be examined.

We have now repeated multiple times the binding of CRIg to activated T cells at different time-points of T cell activation and different concentrations of CRIg-Ig. These new data confirmed our conclusion of that CRIg can bind to activated T cells. Consistent with the conclusion in Figure 2, it showed that the expression of putative CRIg receptors was higher at early stage of T cell activation (the binding intensities: 12h≅24h>>72h). We have now replaced the original Figure 2 with these new data.

We apologize for any confusion in describing the expression of CRIg. CRIg is not expressed in T cells of any subset, or location. We have now included one panel of FACS analyses of CRIg expression in T cells, B cells and NK cells as Figure 1—figure supplement 1.

5) With CRIg^-/-^BL6 mice, are the findings in NOD mice also the case in BL6 mice? Since macrophages from various tissues express CRIg, the authors should examine whether CRIg^-/-^mice have reduced frequencies of Tregs in mucosal tissues such as the gut, peritoneum, skin, and lung, where CRIg^+^tissue macrophages are present.

In contrast to the acceleration of autoimmunity by CRIg deficiency in NOD strain, CRIg^-/-^BL6 mice do not exhibit obvious sign of autoimmune pathology. We analyzed the percentages of Treg cells in small and large intestinal lamina propria (there is no T cells in the pancreas of B6 background) of 6-10 wks old CRIg^-/-^BL6 mice and wildtype littermate controls. Treg percentages were not affected in CRIg^-/-^mice at this age. This is to some extent reminiscent of the phenotype of PD-1^-/-^ mice, as it has been reported that PD-1 deficiency on B6 or Balb/c background only exhibited mild immunopathology only in aged mice (Nishimura H, *Immunity.* 1999. 11:141-51). However, the autoimmune pathology was markedly exacerbated after crossing to autoimmune susceptible strains (Yoshida T, PNAS 2008. 105:3533).

In NOD mice, we have now examined the percentages of pTreg cells in mucosal tissues per the reviewer’s suggestion. Please refer to the answers to comment #2.

6) The rationale for the CRIg/C3 blockade experiments in Figure 2 supplements is unclear. A combination of CRIg-Ig and anti-CRIg completely blocked the proliferation of T conv cells. Although the authors assume that anti-CRIg can block the binding of CRIg to C3 components, this unexplained strong effect of anti-CRIg on T cell suppression is a bit odd. Are active C3 components present in this culture medium? Why does anti-CRIg induce a sort of super agonistic effect of CRIg? On the other hand, this complete T cell proliferation arrest (Figure 2—figure supplement 1) with CRIg and anti-CRIg contradicts with other results (Figure 6—figure supplement 1) in which T cells robustly proliferate in the presence of CRIg and anti-CRIg.

Why CRIg-Ig/anti-CRIg complex exhibited an enhanced effect? Under in vitro settings, we propose that this is an effect of cross-linking. This notion was supported by several observations: soluble CRIg-Ig did not affect T cell proliferation; a plate-bound condition is needed for CRIg-Ig to exhibit a suppressive effect on T cells. Adding anti-CRIg mAb probably further affected the conformational change of CRIg to enhance its engagement with the putative CRIg receptor. To further evaluate the possible cross-linking effect, we have performed a new experiment by coating anti-CRIg (or isotype) mAb to culture plate and adding soluble CRIg-Ig to in vitro cultured CD4 Tconv cells. Again, soluble CRIg-Ig did not suppress T cells if the plate was coated with isotype control mAb. However, in the presence of plate-bound anti-CRIg mAb, soluble CRIg-Ig now can robustly suppress T cell proliferation, further supporting a notion of cross-linking. We have now included these data as Figure 2—figure supplement 2.

Under in vivo settings, we asked whether anti-CRIg/CRIg-Ig complex exhibited a longer serum half-life compared with CRIg-Ig alone. To test this, we treated mice with one dose of CRIg-Ig, or equal amount of CRIg-Ig plus anti-CRIg Ab. The mice were sequentially bled 1, 3, and 7d after injections, and serum concentrations of CRIg were measured by ELISA using Abs of different isotypes and recognizing different epitopes. Clearly, it showed that the inclusion of anti-CRIg maintained higher serum concentrations of CRIg. We have now included these data as Figure 6—figure supplement 2.

Regarding complements in the culture medium. We used heat-inactivated serum for all cell cultures. However, it may still contain complement components, likely from bovine serum (we could not find an ELISA kit to specifically detect bovine complements). Therefore, we chose to use serum-free medium to repeat T cell proliferation and iTreg induction experiments with or without CRIg. However, deprivation of serum for longer than 24 hrs severely affected the viability of cultured T cells, making it difficult to assess the effect of CRIg. We then alternatively conducted a short-term assay in serum-free medium. Since we have demonstrated that CRIg attenuated TCR activation signaling cascades by measuring the phosphorylation of ZAP70, Erk1/2, AKT and S6. Therefore, using the same approach, except using serum-free culture medium. Indeed, CRIg-Ig could attenuate TCR activation without complements. We have now included these data in Figure 2—figure supplement 1.

Re the difference between Figure 2—figure supplement 1 and Figure 6—figure supplement 1.That was due to the presence of TGF- b in Figure 6—figure supplement 1. TGF-β interferes the suppressive effect of CRIg-Ig on T cells.

7) In Figure 3, the differentiation of pTregs from Tconv was impaired in the pancreas, but not in LNs of CRIg^-/-^NOD mice which had received adoptive transfer of BDC2.5 Tconv. This is a bit odd. It is unlikely that there is enough cellular niche to allow Tconv to get activated, expand, and differentiate into pTreg in T cell-sufficient NOD mice. They should show more detailed FACS profiles, for example, the proportion of Thy1.1 (transferred) vs Thy1.2 (resident) in Foxp3^+^ Tregs in the pancreas and LNs.Also, CRIg-Ig treatment enhanced pTreg generation in pancreas but not in panLNs and ILNs in Figure 3. The authors should discuss the reason for this result.

For Figure 3 (now became Figure 3), in these experiments, we chose 4-wk old recipient, a time point at which insulitis just started. We used activated TCR monoclonal BDC2.5 T cells that were highly diabetogenic and had the advantages for competition with polyclonal endogenous T cells. We have now added FACS data to show the engraftments of transferred Thy1.1^+^ T cells and the fractions of them becoming GFP(Foxp3)^+^ (Figure 3—figure supplement 2). Because the key point here was how much transferred Tconv cells were converted into Treg cells in the presence or absence of CRIg-Ig, we only analyzed transferred cells, not endogenous Treg cells.

Re “Why CRIg-Ig treatment enhanced pTreg generation in pancreas but not in panLNs and ILNs?”This is mainly due to that pTreg cells are primarily differentiated at tissue sites. Mice deficient for Foxp3 conserved noncoding sequence 1 (CNS1) have normal numbers of tTregs but have severe deficit of pTregs (Zheng et al., 2010). The defects are most striking at mucosal surfaces, which are the primary sites for pTreg generation. We have strengthened this statement in the text.

8) Is the target molecule of CRIg on effector T cells and regulatory T cells same or different? The authors should discuss how CRIg exhibit opposite effects on the two different T cell populations.

Without knowing the cognate receptor for CRIg in Tconv or Treg cells, we here provide our postulations based on the data from the binding and functional assays. First, CRIg exhibits a coinhibitory effect on T cells. For this function, I don’t think it is opposite between Tconv and Treg cells. CRIg can bind to both Tconv and Treg cells. However, the binding intensity was substantially higher in Tconv than that in Treg cells, suggesting that the putative receptor that mediates coinhibitory effect is more abundantly expressed in activated Tconv than that in Treg cells. Consistent with this, we found that under the same experimental settings CRIg more potently suppressed Tconv than Treg cells (as shown in Author response image 1).

Regarding the effect of CRIg in promoting Treg generation. Our data supported that the putative receptor mediating T cell suppression is not necessarily needed for Treg differentiation by CRIg. This is because that CRIg-Ig would not suppress T cells at the late stage of T cell in vitro activation and proliferation (Figure 2), which is very likely due to the transient expression of the receptor for CRIg in T cells. However, we did a new experiment and found that CRIg can still promote the conversion of these T cells at their late stage of in vitro activation/proliferation into Foxp3^+^ Treg cells in the presence of TGF-β. This suggested that another mechanism is involved in Treg induction. We have now included these data and also revised the discussions accordingly.

9) It is hard to understand why the authors focused on environmental factors that modulate CRIg expression on TRMs in the tissues other than pancreas. Is there any relationship between CRIg levels in pancreatic TRMs and bacterial colonization or RA? How about the expression of CRIg in pancreatic TRMs after birth in wild-type and NOD mice? It is more valuable for this study to find the environmental cues that would regulate CRIg levels in pancreatic TRMs, if possible, especially during the development of type 1 diabetes.

We thank the reviewer for these comments. Related to comment #12 below, we have now performed new experiments to dissect the connections between environmental factors (gut microbiota and RA) and CRIg^+^ TRMs in the pancreas especially during the development of T1D, as the reviewer pointed out. We have included the new data as Figure 7 and Figure 7. Interestingly, while antibiotic treatment diminished pancreatic CRIg^+^ TRMs, blocking RA signaling in vivofailed to do so, suggesting different regulations were utilized at different tissues to maintain CRIg expression in TRMs. A recently study demonstrated that antibiotic-mediated gut microbiome perturbation accelerates development of type 1 diabetes in mice (Livanos et al., 2016). The dampened expression of CRIg in pancreatic TRMs could be one possible mechanism.

10) Does CRIg bind to Treg and synergize with TGFb? Binding of CRIg to Tregs should be added (as in Figure 2 for activated T cells) along with key controls.

The promoting effect of CRIg-Ig in iTreg differentiation could be directly through enhancing TGF-β signaling or potentiating other signaling in iTregs independent of TGF- b. We have now conducted the phosphorylation assays of Smad2/3 in iTreg condition in the presence or absence of CRIg-Ig and found that CRIg-Ig did not further enhance TGF- b induced Smad2/3 phosphorylation. We have previously found that CRIg-Ig suppress the activities of AKT and S6 in T cells under iTreg condition. Therefore, CRIg-Ig does not directly affect TGF-b signaling, instead, potentiates other pathways that facilitate iTreg differentiation. We have now added the Smad2/3 phosphorylation results as Figure 3—figure supplement 1.

We have now demonstrated that CRIg can bind to activated Treg cells, but to a much lower level compared to that in Tconv cells. We have now included these data as Figure 2—figure supplement 3.

11) Treg stability depends on the methylation status of FoxP3 CNS2 region. The authors have shown that CRIg does not affect the methylation status of CNS2 site of Foxp3 yet FoxP3 expression in adoptively transferred iTregs generated with CRIg-Ig is remarkably stable. The inference from this observation is that Tregs do not need continuous cell:cell contact with CRIg-expressing macrophages to maintain FoxP3 expression and function. The authors should discuss this point.

We thank the reviewer for this comment. Our data supported this inference. As we have found that transferred iTreg cells preconditioned by CRIg were more stable even in spleen, where there was no CRIg^+^ cells. We have further strengthened this point in the Results and Discussion.

12) The experiments in Figure 6 and Figure 7 are interesting and compelling. However they do not connect well with the rest of the paper. These features should have been examined in pancreatic macrophages to make Figure 6 and Figure 7 cohesive with Figure 1-5.

Please see the answers to comment #9.